# Unc-4 acts to promote neuronal identity and development of the take-off circuit in the *Drosophila* CNS

Haluk Lacin[1,2]*, W Ryan Williamson[1], Gwyneth M Card[1], James B Skeath[2], James W Truman[1,3]

[1]Janelia Research Campus, Howard Hughes Medical Institute, Ashburn, United States; [2]Department of Genetics, Washington University, Saint Louis, United States; [3]Friday Harbor Laboratories, University of Washington, Friday Harbor, United States

**Abstract** The *Drosophila* ventral nerve cord (VNC) is composed of thousands of neurons born from a set of individually identifiable stem cells. The VNC harbors neuronal circuits required to execute key behaviors, such as flying and walking. Leveraging the lineage-based functional organization of the VNC, we investigated the developmental and molecular basis of behavior by focusing on lineage-specific functions of the homeodomain transcription factor, Unc-4. We found that Unc-4 functions in lineage 11A to promote cholinergic neurotransmitter identity and suppress the GABA fate. In lineage 7B, Unc-4 promotes proper neuronal projections to the leg neuropil and a specific flight-related take-off behavior. We also uncovered that Unc-4 acts peripherally to promote proprioceptive sensory organ development and the execution of specific leg-related behaviors. Through time-dependent conditional knock-out of Unc-4, we found that its function is required during development, but not in the adult, to regulate the above events.

*For correspondence:
lacinhaluk@gmail.com

**Competing interests:** The authors declare that no competing interests exist.

## Introduction

How does a complex nervous system arise during development? Millions to billions of neurons, each one essentially unique, precisely interconnect to create a functional central nervous system (CNS) that drives animal behavior (*Herculano-Houzel, 2009*). Work over several decades shows that developmentally established layers of spatial and temporal organization underlie the genesis of a complex CNS. For example, during spinal cord development in vertebrates, different types of progenitor cells arise across the dorso-ventral axis and generate distinct neuronal lineages in a precise spatial and temporal order. The pMN progenitors are located in a narrow layer in the ventral spinal cord and generate all motor neurons (*Tsuchida et al., 1994*; *Pfaff et al., 1996*; *Briscoe et al., 2000*). Similarly, twelve distinct pools of progenitors that arise in distinct dorso-ventral domains generate at least 22 distinct interneuronal lineages. Within each lineage, neurons appear to acquire similar identities: they express similar sets of transcription factors, use the same neurotransmitter, extend processes in a similar manner and participate in circuits executing a specific behavior (*Jessell, 2000*; *Alaynick et al., 2011*; *Lu et al., 2015*).

The adult *Drosophila* ventral nerve cord (VNC), like the vertebrate spinal cord, also manifests a lineage-based organization. The cellular complexity of the VNC arises from a set of segmentally repeated set of 30 paired and one unpaired neural stem cells (Neuroblasts [NBs]), which arise at stereotypic locations during early development (*Doe, 1992*). These individually identifiable NBs undergo two major phases of proliferation: the embryonic phase generates the functional neurons of the larval CNS, some of which are remodeled to function in the adult, and the post-embryonic phase generates most of the adult neurons (*Truman and Bate, 1988*; *Truman, 1990*; *Prokop and*

*Technau, 1991*; *Bossing et al., 1996*; *Schmid et al., 1999*; *Consoulas et al., 2002*). The division mode within NB lineages adds another layer to the lineage-based organization of the VNC. Each NB generates a secondary precursor cell, which divides via Notch-mediated asymmetric cell division to generate two neurons with distinct identities (*Spana and Doe, 1996*; *Skeath and Doe, 1998*). After many rounds of such cell divisions, each NB ends up producing two distinct hemilineages of neurons, termed Notch-ON or the 'A' and Notch-OFF or the 'B' hemilineage. In this paper, we focus only on postembryonic hemilineages, which from this point on in the paper we refer to as hemilineages for simplicity. Within a hemilineage, neurons acquire similar fates based on transcription factor expression, neurotransmitter usage, and axonal projection (*Truman et al., 2010*; *Lacin et al., 2014*; *Lacin et al., 2019*). Moreover, neurons of each hemilineage appear dedicated for specific behaviors. For example, artificial neuronal activation of the glutamatergic hemilineage 2A neurons elicit specifically high frequency wing beating, while the same treatment of the cholinergic hemilineage 7B neurons leads to a specific take-off behavior (*Harris et al., 2015*). Thus, hemilineages represent the fundamental developmental and functional unit of the VNC.

We previously mapped the embryonic origin, axonal projection pattern, transcription factor expression, and neurotransmitter usage of essentially all hemilineages in the adult *Drosophila* VNC (*Truman et al., 2004*; *Truman et al., 2010*; *Lacin et al., 2014*; *Harris et al., 2015*; *Lacin and Truman, 2016*; *Shepherd et al., 2016*; *Lacin et al., 2019*; *Shepherd et al., 2019*). Here, we leverage this information to elucidate how a specific transcription factor, Unc-4, acts within individual hemilineages during adult nervous system development to regulate neuronal connectivity and function, and animal behavior. Unc-4, an evolutionarily conserved transcriptional repressor, is expressed postmitotically in seven of the 14 cholinergic hemilineages in the VNC: three 'A' -Notch-ON- hemilineages (11A, 12A, and 17A) and four 'B' -Notch-OFF- hemilineages (7B, 18B, 19B, and 23B) (*Miller et al., 1992*; *Rovescalli et al., 1996*; *Lacin et al., 2014*; *Lacin et al., 2019*; *Nittoli et al., 2019*). For four of the Unc-4$^+$ hemilineages (7B, 17A, 18B, and 23B), the neurons of the sibling hemilineage undergo cell death (*Truman et al., 2010*). For the remaining three (11A, 12A, and 19B), the neurons of the sibling hemilineage are GABAergic (*Lacin et al., 2019*). Unc-4 expression in these hemilineages is restricted to postmitotic neurons and it appears to mark uniformly all neurons within a hemilineage during development and adult life (*Lacin et al., 2014*; *Lacin and Truman, 2016*).

We generated a set of precise genetic tools that allowed us to uncover lineage-specific functions for Unc-4: in the 11A hemilineage, Unc-4 drives the cholinergic identity and suppresses the GABAergic fate; in the 7B hemilineage, Unc-4 promotes correct axonal projection patterns and the ability of flies to execute a stereotyped flight take-off behavior. We also find that Unc-4 is expressed in the precursors of chordotonal sensory neurons and required for the development of these sensory organs, with functional data indicating Unc-4 functions in this lineage to promote climbing, walking, and grooming activities.

## Results

In the absence of extant mutant alleles of *unc-4* gene, we used CRISPR/Cas9 technology to engineer both null and conditional null alleles of *unc-4* as well as GAL4 and split-GAL4 reporter lines of *unc-4* in order to assess *unc-4* function in the postembryonic CNS (*Figure 1*; see also Materials and methods). First, we generated the *unc-4*$^{FRT}$ fly line in which we inserted two Flippase Recognition Target (FRT) sites in the same orientation flanking the 2$^{nd}$ and 3$^{rd}$ *unc-4* exons as well as an attP site immediately 5' to the second exon. We used this parental *unc-4*$^{FRT}$ stock to generate all mutant lines and the GAL4/split GAL4 reporter lines. To generate the *unc-4*$^{null}$ mutant line, we used Flippase (FLP) to excise the 2$^{nd}$ and 3$^{rd}$ exons of *unc-4* in the germline, generating an *unc-4* allele that lacks the homeodomain (*Figure 1A,B*). Molecular analysis confirmed the presence of the expected deletion and immunostainings revealed no detectable protein expression in the *unc-4*$^{null}$ mutant (*Figure 1C*).

To generate GAL4 and split GAL4 lines reporter lines of *unc-4*, we used Trojan-GAL4 system (*Diao et al., 2015*) to insert three different GAL4 exons – GAL4, GAL4-AD, and GAL4-DBD into the attP site of the *unc-4*$^{FRT}$ line (DBD: DNA binding domain of GAL4; AD: activation domain of GAL4). When crossed to appropriate transgenes, each of these alleles faithfully recapitulated Unc-4 expression in the VNC and brain in the larval and adult CNS (*Figure 1D–F* and not shown). Each of these lines is also a null allele of *unc-4*, as the Trojan exons truncate the Unc-4 protein after its first exon.

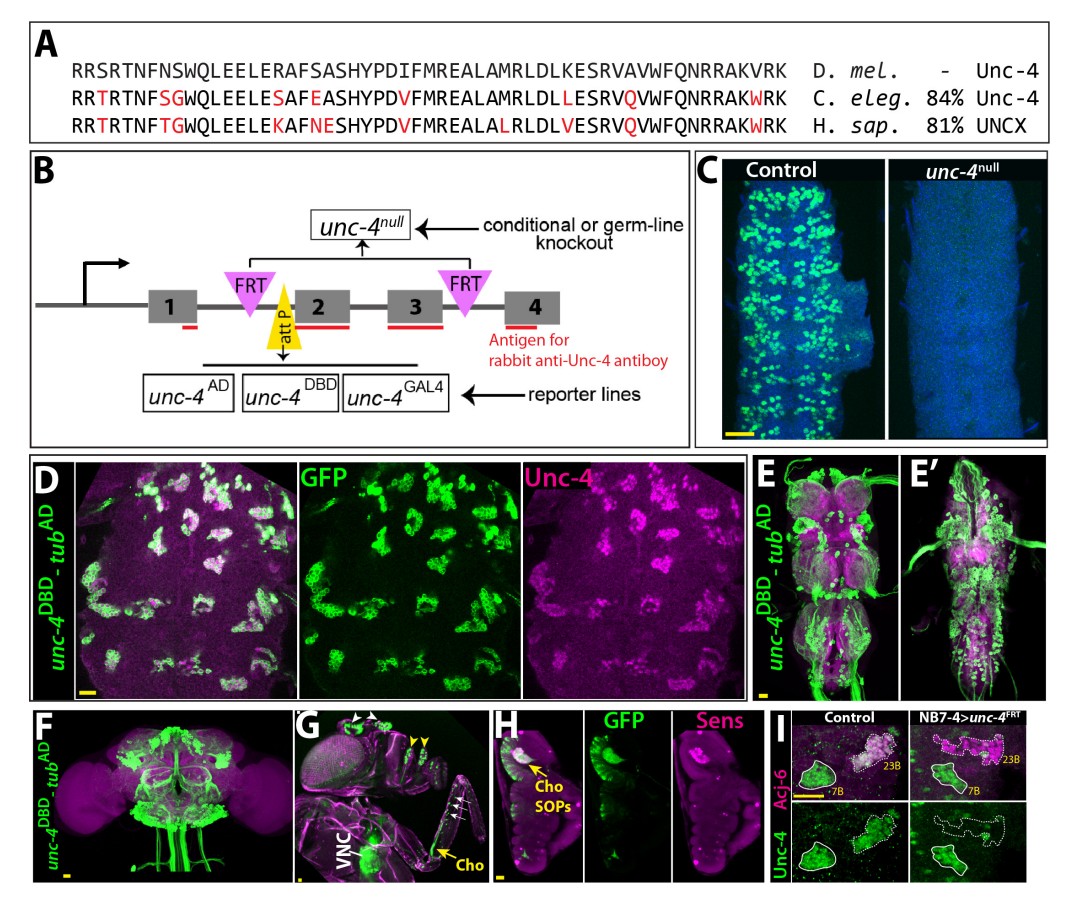

**Figure 1.** Tools generated to study Unc-4 function. (**A**) Homeodomains of Unc-4 protein in *D. melanogaster*, *C. elegans* and *H. sapiens* are shown with the percent identity on the right. Amino acids that differ from *Drosophila* sequence were highlighted in red. (**B**) Schematic representation of edited *unc-4* locus in *unc-4*^FRT line shown (drawing is not to scale). Two FRT sites (magenta) are present with the same orientation flanking 2nd and 3rd exons (the homeodomain coding exons) and an attP site (yellow) is located 5′ to the second exon. *unc-4*^null fly line was generated by FLP/FRT mediated excision of these exons in the germline. The same FLP/FRT mediated excision in somatic tissues was used for conditional mutant analysis. Trojan exons were inserted into the attP site to generate *unc-4*^AD, *unc-4*^DBD, *unc-4*^GAL4 reporter lines. The region that is recognized by rabbit Unc-4 antibody underlined in red. (**C**) Dissected VNC samples from control (left) and *unc-4*^null (right) embryos are shown. In the control VNC, Unc-4 protein (green) is expressed in a segmentally-repeated pattern. In the mutant embryo, no Unc-4 expression is detected. (**D–H**) *unc-4*^DBD-*tub*^AD driven GFP expression (green) shown. (**D**) Expression of this split GAL4 combination overlaps with Unc-4 protein expression (magenta) in the larval VNC. The clusters of neurons correspond to ventrally located Unc-4+ hemilineages. (**E–F**) Unc-4 is expressed in clusters of neurons in the adult VNC (**E, E′**) and brain (**F**). Ventrally located VNC lineages shown in E; dorsally located VNC lineages shown in E′. Here and in other figures, CadN antibody staining (magenta) used to define the contours of the adult CNS tissues. (**G**) Anterior body parts of an adult fly shown. Unc-4 reporter marks different types of sensory neurons: chordotonal (cho) organ (yellow arrow) and bristle sensory (white arrows) neurons in the leg, olfactory neurons in maxillary pulps (yellow arrowheads) and antennas (white arrowheads), and Johnston's organ (not visible in this image). Magenta is cuticle autofluorescence. (**H**) A leg disc from an early stage pupa shown; anterior is to the left. Unc-4 is expressed in progenitors of chordotonal neurons, also called sensory organ precursors (SOPs), which were marked with Sens expression (magenta). (**I**) An example for conditional removal of *unc-4* shown. Unc-4 (green) and Acj6 (magenta) are co-expressed in 23B neurons in the larval VNC but only Unc-4 is expressed in 7B neurons. NB7-4 specific FLP expression in the *unc-4*^FRT larva (right panel) removes Unc-4 expression from the 23B neurons (NB7-4 progeny) but not the 7B neurons. Anterior is up in all images in this and other figures unless indicated otherwise. Scale bar (yellow line) is 20 microns.

The online version of this article includes the following figure supplement(s) for figure 1:

**Figure supplement 1.** Unc-4 expression pattern in larval and adult tissues.

This conclusion is confirmed by the absence of detectable levels of Unc-4 protein in larvae hemizygous for *unc-4*^GAL4, *unc-4*^AD, and *unc-4*^DBD (not shown). Leveraging these transgenes, we tracked Unc-4 expression in the peripheral nervous system (PNS) and found that Unc-4 is expressed in all progenitors of leg chordotonal neurons (also called sensory organ precursors [SOPs]) and head sense

organs in the larvae as well as many adult sensory neurons including chordotonal and bristle sensory neurons in the leg and Johnston's organ and olfactory neurons in the antenna (*Figure 1G–H*; *Figure 1—figure supplement 1*).

To determine if we can use the *unc-4*[FRT] line to ablate Unc-4 function in a lineage-specific manner, we used NB7-4 -GAL4 to drive UAS-linked FLP expression specifically in NB7-4 during embryonic stages in *unc-4*[FRT] homozygotes (*Lacin and Truman, 2016*). NB7-4 gives rise to the Unc-4[+] 23B neurons (*Lacin et al., 2014*). This manipulation resulted in loss of Unc-4 expression specifically in 23B neurons, but no other lineage (*Figure 1I*), highlighting the ability of our system to ablate Unc-4 function in a lineage specific manner.

## Loss of Unc-4 results in defined behavioral defects

All loss of function *unc-4* alleles we generated (null, AD, DBD, and GAL4) exhibited semi-lethality in the hemizygous or homozygous state, with many pharate adults failing to eclose from the pupal case. Escaper *unc-4* null flies emerged as adults with a frequency of 13% of the expected Mendelian rate. Adult *unc-4* null mutant males were fertile, but females were infertile. Examination of *unc-4* females revealed that they were impaired in egg laying and their ovaries were dramatically enlarged due to egg retention (not shown), a phenotype that may arise due to defective functioning of sensory neurons in the female reproductive tract, some of which express Unc-4 (*Figure 1—figure supplement 1*). *unc-4* mutant adults also exhibited defects in VNC executed behaviors (*Figure 2*). All *unc-4* mutant animals manifested erect wings, and none could fly or jump (*Figure 2A,C,I*; *Video 1*). *unc-4* mutant adults could walk but only in an uncoordinated manner as most failed a climbing assay (see methods) due to falling or being extremely slow (*Figure 2G,I*; *Videos 2* and *3*). *unc-4* mutants also showed impaired grooming due to lack of coordination among legs, and they completely failed in executing three-leg rubbing (*Figure 2E,I*; *Video 4*), a characteristic behavior which requires coordination of three legs for cleaning (*Phillis et al., 1993*). *unc-4*[DBD]-*tub*[AD] driven *unc-4* expression rescued the observed behavioral defects as well as the semilethality (*Figure 2B,D,F,H*; *Videos 2–*

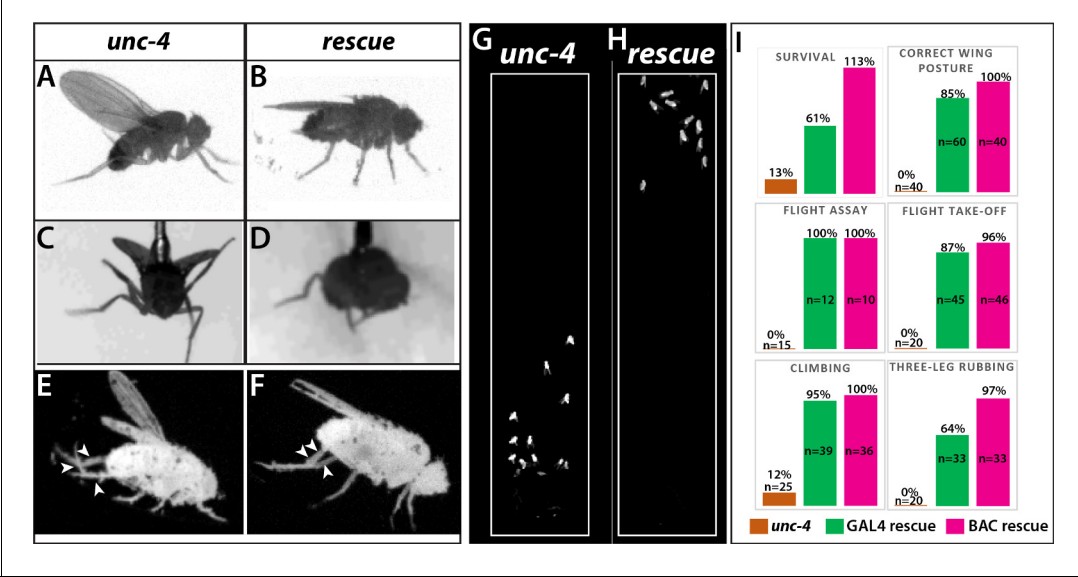

**Figure 2.** Loss of Unc-4 function results in behavioral defects in adult flies. (**A–H**) Still images from recorded videos showing the behavioral defects observed in *unc-4* mutants (A, C, E, and G) and the rescue of these defects with *unc*[DBD]- *tub*[AD] driven UAS-*unc-4* transgene (B, D, F, and H). Images from control animals not shown as they were virtually identical to the images from rescued animals. Wings of *unc-4* mutants are locked in an erect position (**A**); they fail flying in a tethered flight assay (**C**). Restoring Unc-4 function enables mutant animals to rest their wings in tucked position (**B**) and fly in the flight assay (**D**). Mutant animals fail to bring their three legs together for rubbing off dust (**E**) and rescue animals regain this three-leg rubbing behavior (**F**). (**G–H**) Restoring Unc-4 expression also improves climbing defects of *unc-4* mutants. Positions of mutant and rescued animals after bang-induced climbing shown; the image was reproduced from *Video 5*. (**I**) Percentage bar graphs showing the quantification of *unc-4* mutant phenotypes (orange bars) and their rescue with *unc*[DBD]- *tub*[AD] driven UAS-*unc-4* (green bars) or a BAC transgene (PBac(DC335)) containing the *unc-4* locus (magenta bars). See methods for detailed explanations for the assays quantified here. See also *Videos 1–7*.

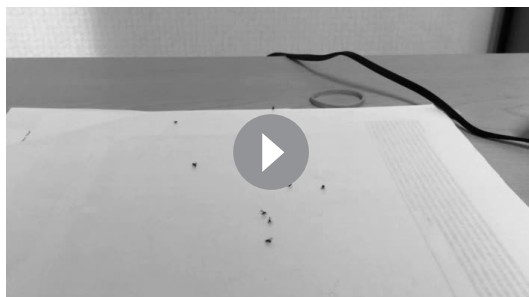

**Video 1.** *unc-4* mutant males (*unc-4*^DBD /y) in open field.
https://elifesciences.org/articles/55007#video1

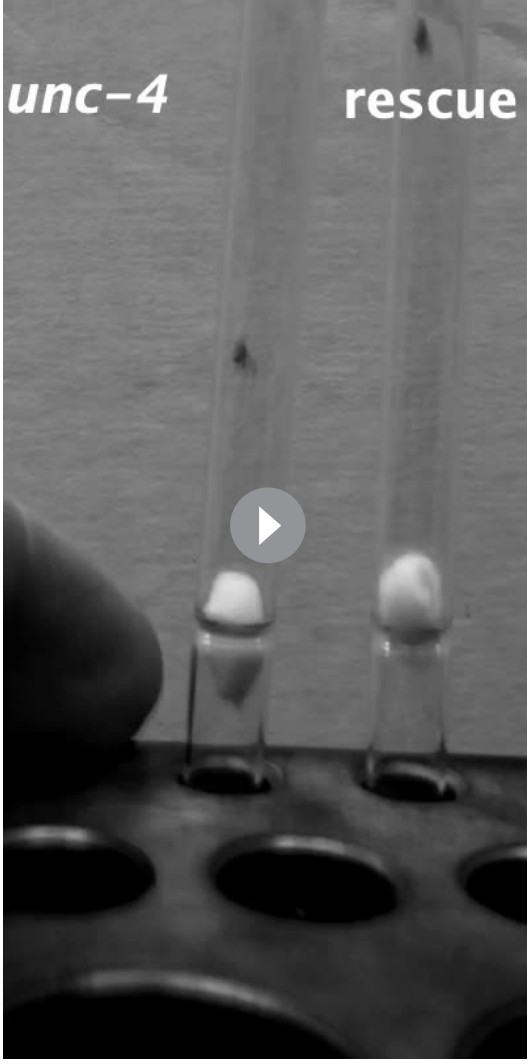

**Video 2.** Climbing activity of individual flies: left, *unc-4* mutant males (*unc-4*^DBD /y); right, rescued males (*unc*^DBD/y; *tub*^AD/10XUAS-IVS-myr::GFP; UAS-*unc-4*). https://elifesciences.org/articles/55007#video2

*6*). These rescued animals, however, did not behave like wild type: they were still slow to take-off (*Video 6*) and still exhibited defects in three-leg rubbing (not shown). To ensure that all observed defects were the result of loss of Unc-4 function, we introduced a BAC transgene that contains the *unc-4* locus (*Venken et al., 2010*) into *unc-4* mutant flies and observed essentially 100% rescue of all observed phenotypes (*Figure 2I*; also compare *Videos 6* and *7*). Our findings, thus, clearly show that Unc-4 function is required for take-off, flight, climbing and grooming behaviors.

## Unc-4 function is required during development

Unc-4 is expressed in both the developing and adult nervous system. To determine if *unc-4* related behavioral defects arise due to a role for Unc-4 during development or a later role for Unc-4 during adult life, we used the TARGET system (*McGuire et al., 2004*) to remove Unc-4 function at different time points by controlling the timing of FLP expression. We removed Unc-4 function starting from the embryo or from late pupal stages and assessed the resulting flies after two weeks (see the methods) for the erect wing, three-leg rubbing and climbing phenotypes. We observed that early embryonic removal of Unc-4 resulted in the same behavioral defects as seen in *unc-4* mutants, but its removal in late pupal stages yielded flies that behaved like wild-type (not shown). Unc-4, therefore, functions during development to regulate this suite of specific behaviors.

## Peripheral contributions to the *unc-4* behavioral phenotypes

Next, we wanted to determine whether Unc-4 functions within specific CNS or PNS lineages to control specific behaviors. We used two GAL4 driver lines in conjunction with our *unc4*^FRT line to remove Unc-4 function in both the CNS and PNS or just the CNS. We used *sca*-GAL4 to remove Unc-4 function in both the CNS and PNS (*Mlodzik et al., 1990*) and *dpn*^EE- GAL4 to remove Unc-4 function only in the CNS (*Yang et al., 2016*; *Figure 3A–H*). *sca*-GAL4 removed most Unc-4 expression from the CNS, all of Unc-4 expression from the progenitors of chordotonal sense organs, and also Unc-4 expression in epithelial cells of the leg disc (*Figure 3B,F*). *dpn*^EE-GAL4 which is expressed specifically in CNS NBs (*Emery and Bier, 1995*; *Yang et al., 2016*) removed most Unc-4 expression from the

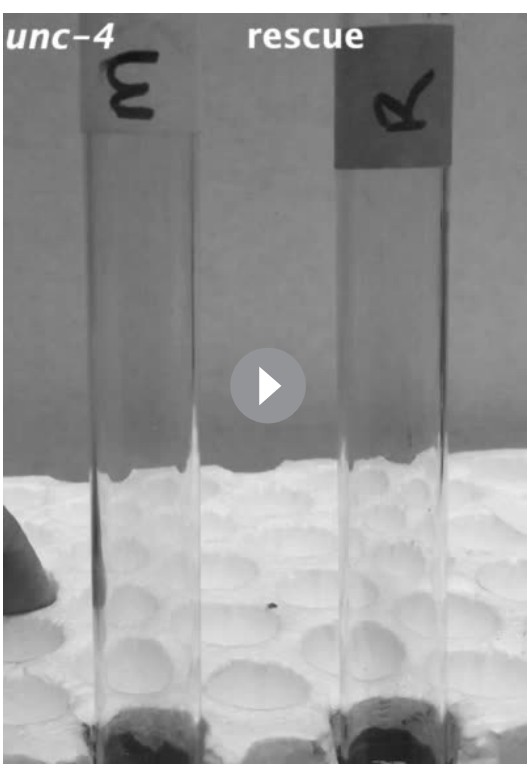

**Video 3.** Climbing activity of group of flies: left, *unc-4* mutant males (*unc-4*DBD /y); right, rescued males (*unc*DBD/y; *tub*AD/10XUAS-IVS-myr::GFP; UAS-*unc-4*).
https://elifesciences.org/articles/55007#video3

CNS but left Unc-4 expression in leg discs largely intact (*Figure 3C,G*). *sca*-GAL4 mediated knock-out of Unc-4 function led to the same behavioral defects observed in *unc-4* mutant animals: all had erect wings, failed to fly, and showed grooming and climbing defects (*Table 1*; *Video 8*). By contrast, the *dpn*EE-GAL4 mediated removal from the CNS gave a partial phenotype with the flies having erect wings, an inability to fly, and mild grooming defects (*Table 1*), but they performed well on the climbing assay scoring at the same level as controls (*Table 1*; *Video 9*). These results suggest that Unc-4 related PNS deficits are responsible for the poor climbing performance, while both PNS and CNS deficits are responsible for grooming defects.

Due to the known role of the chordotonal organs in leg-related behaviors (*Tuthill and Wilson, 2016*), we asked if Unc-4 function in the chordotonal neurons is required for wild-type climbing behavior. To do this, we used a chordotonal SOP-specific driver *ato*-GAL4 (*Hassan et al., 2000*) to remove Unc-4 function completely from chordotonal lineage (progenitors and neurons), while leaving Unc-4 expression in the CNS intact (*Figure 3D,H*). *ato*-GAL4 also unexpectedly removed Unc-4 expression in the epithelial cells of most leg discs (*Figure 3H*). *ato*-GAL4 mediated removal of Unc-4 function in chordotonal lineage and epithelial cells resulted in coordination defects in leg movements: we observed defects in three-leg rubbing (*Table 1*; *Video 10*) and walking (*Videos 11* and *12*). We did not assess climbing behavior of the resulting flies because *ato*-GAL4 driven FLP expression impaired the negative-geotaxis behavior of control animals (presumably due to *ato*-GAL4 expression in the gravity sensing Johnston organ or its progenitor cells). To assess walking, we challenged flies to walk horizontally in small-diameter tubes. Control animals walked in a coordinated manner without falling, but *unc-4* null mutants and *ato*-GAL4 >*unc4* FRT flies were slow and fell often (*Videos 11* and *12*; *Figure 3I*). Thus, our results indicate that loss of *unc-4* function in the leg discs, presumably in the chordotonal lineage, is required to drive proper leg-related behaviors, including three-leg rubbing, walking, and likely climbing.

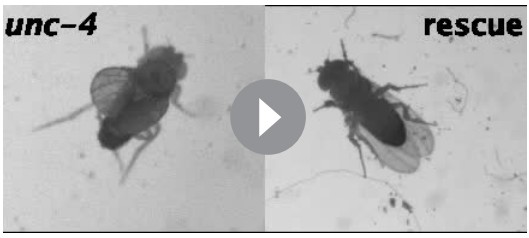

**Video 4.** Three-leg rubbing activity of individual flies; 2X slower: left, *unc-4* mutant males (*unc-4*DBD /y); right, rescued males (*unc*DBD/y; *tub*AD/10XUAS-IVS-myr::GFP; UAS-*unc-4*).
https://elifesciences.org/articles/55007#video4

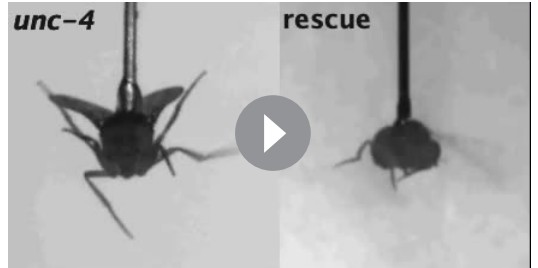

**Video 5.** Tethered flight assay, 10X slower: left, *unc-4* mutant males (*unc-4*DBD /y); right, rescued males (*unc*DBD/y; *tub*AD/10XUAS-IVS-myr::GFP; UAS-*unc-4*).
https://elifesciences.org/articles/55007#video5

To ascertain whether Unc-4 function during chordotonal organ development may underlie the observed behavioral defects, we assessed Unc-4 function on adult femoral chordotonal organ (feCO) neurons, the major chordotonal organ in the leg. We visualized them in control and *unc-4* mutant animals via *iav*-GAL4, which marks all chordotonal neurons in the adult (*Kwon et al., 2010*). In control animals, feCO neuronal cell bodies are clustered and located proximally near the trochanter border; they extend dendrites in a stereotyped fashion (*Figure 3J*). In the mutants, the cell bodies of many of the neurons were displaced across the femur (*Figure 3K,L*) and their bundled dendritic projections appeared to be disrupted as compared to controls; all of these defects were res-

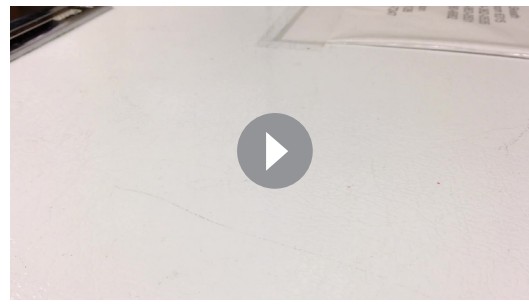

**Video 6.** Rescue of the flight take-off activity by GAL4/UAS system (*unc*DBD/y; *tub*AD/10XUAS-IVS-myr::GFP; UAS-*unc-4*).
https://elifesciences.org/articles/55007#video6

cued with restored Unc-4 function (*Figure 3M*). Of note, axonal projections of the FeCO neurons in the VNC of *unc-4* mutant animals appeared like wild-type (not shown). Unc-4, therefore, appears to regulate the patterning and dendritic projections of chordotonal neurons in the leg, with this function likely underlying the observed defects in three-leg rubbing, walking, and climbing.

## Unc-4 and neurotransmitter expression

The central expression of Unc-4 is striking because that seven hemilineages that express it are all cholinergic (*Lacin et al., 2019*). To determine if Unc-4 function is required for the survival or proper differentiation of these neurons, we combined *unc-4*DBD line with the ubiquitous *tub*AD line to visualize all Unc-4$^+$ neurons in control and *unc-4* mutant animals. As shown in *Figure 4A*, we observed no gross defects in the number and position of Unc-4$^+$ neurons in the absence of Unc-4 function, indicating Unc-4 is not required for its own expression nor is it required for the survival of these neurons.

We then determined the neurotransmitter profile of these neurons in control and *unc-4* mutant background by intersecting *unc-4* split GAL4 lines with neurotransmitter specific split-GAL4 lines expressed in cholinergic, glutamatergic, or GABAergic neurons (*Diao et al., 2015; Lacin et al., 2019*). We previously showed that all of the Unc-4$^+$ postembryonic hemilineages are cholinergic and that half of all cholinergic hemilineages (7 of 14) express Unc-4 in the VNC (*Lacin et al., 2019*). To confirm this and visualize cholinergic Unc-4$^+$ neurons, we intersected *unc-4*AD expression with the cholinergic neuronal driver *ChAT*DBD (*Figure 4B*). As expected, most Unc-4$^+$ neurons including neurons of all seven postembryonic hemilineages (identified based on location and axonal projection) in the control VNC were marked with the split-GAL4 combination of *unc-4*AD-*ChAT*DBD, confirming the cholinergic identity of the Unc-4$^+$ hemilineages.

In the *unc-4* mutant VNC, the expression of this split-GAL4 combination appeared grossly normal and marked all seven Unc-4$^+$ postembryonic hemilineages, indicating that Unc-4 function is not necessary for the cholinergic fate of most (but not all, see below) neurons in these hemilineages.

The relationship of Unc-4 expression to the glutamatergic phenotype was determined by combining *VGlut*DBD, which labels glutamatergic neurons, with *unc-4*AD. Here, *unc-4*AD-*VGlut*DBD driven GFP expression identified several glutamatergic Unc-4$^+$ motor neurons in thoracic and abdominal segments in addition to a few occasional interneurons in the thoracic segments (*Figure 4C*). Based on their axon projections,

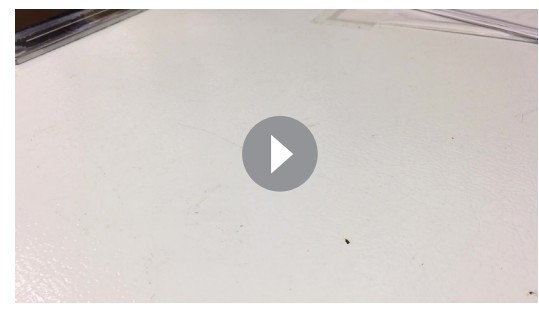

**Video 7.** Rescue of the flight take-off activity by a BAC transgene (*unc*DBD/y;; PBac(DC335) /+).
https://elifesciences.org/articles/55007#video7

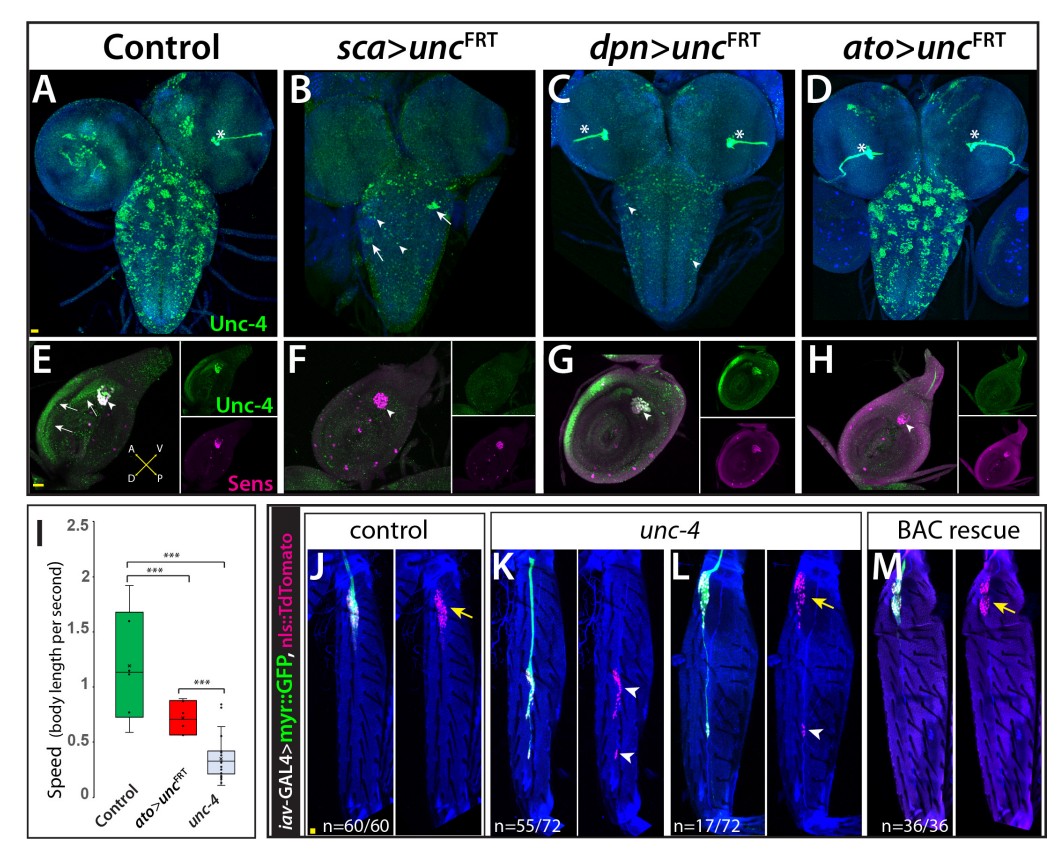

**Figure 3.** Unc-4 function in the periphery is required for animal coordination and the formation of chordotonal sensory organ. Unc-4 expression (green) in the CNS (A–D) and leg disc (E–H) of wandering-stage larvae shown. *sca*-GAL4 (B, F), *dpn*^EE^-GAL4 (C, G), *ato*-GAL4 (D, H) driven FLP expression in the *unc-4*^FRT^ line used to remove Unc-4 function in a tissue-specific manner. (A) In the control CNS, Unc-4 is expressed in clusters of post-embryonically born neurons (~50 clusters per CNS). (B) Manipulation with *sca*-GAL4 removed Unc-4 expression in most of the neuronal clusters but spared a few clusters stochastically (arrows; 3.4 clusters per CNS, N = 5). (C) *dpn*^EE^-GAL4 removed Unc-4 expression in all clusters without sparing any cluster (N = 5). In both *sca*-GAL4 and *dpn*^EE^-GAL4 manipulations, some dorsally located neurons (presumably embryonic-born) retained Unc-4 expression (arrowheads). (D) *ato*-GAL4 manipulation did not appear to affect Unc-4 expression in the CNS. Asterisks in (A–D) indicates the bleed-through from the 3xP-DsRed transgene present in *unc-4*^FRT^ line; thus, they do not represent Unc-4 expression. (E–H) Unc-4 (green) and Sens (magenta) expression in leg discs shown. (E) In the control leg disc, Unc-4 is expressed in precursors of the chordotonal organ (arrowhead, marked with Sens) as well as in epithelial cells in the anterior compartment (arrows). (F) *sca*-GAL4 removed Unc-4 expression in all SOPs (16/16 leg discs, N = 5 animals), and also from the epithelial cells completely (10/16) or partially (6/16; not shown). (G) *dpn*^EE^-GAL4 manipulation did not affect Unc-4 expression in most leg discs (15/23, N = 9 animals) but sometimes removed Unc-4 expression in SOPs partially (3/23; not shown) or completely (5/23; not shown). (H) *ato*-GAL4 manipulation removed Unc-4 expression in all SOPs (25/25 leg discs; N = 8 animals) but also in epithelial cells completely (20/25) or partially (5/25; not shown). (I) *ato*-GAL4 mediated removal of Unc-4 resulted in slower walking animals, which were still faster than *unc-4* mutant animals (***$p<0.0001$ student's t test) (see also *Videos 11* and *12*). The walking speed was 1.4 +/- 0.6 body length per second (BLPS) for control , 0.76 +/- 0.4 BLPS for ato-GAL4> *unc4*^FRT^ and 0.35 +/- 0.2 BLPS for *unc-4* mutants (n > 27 tracts from 10 animals each). (J–L) The femur of T2 (J, K, M) and T1 (L) legs shown. *iav*-GAL4 driven myr::GFP (green) and nls::TdTomato (red) used to visualize processes and cell-body location of the chordotonal neurons in the femur. (J) In the control leg, clusters of chordotonal neurons were located proximally and close to the trochanter border in the femur (arrow). (K–L) In the *unc-4* mutant animals, organization of the chordotonal neurons was disrupted. All animals showed this phenotype with varying degree of severity. In many legs (55/72; N = 12 animals), all of the *iav*-GAL4 marked chordotonal neurons were located away from their proximal position (arrowheads in K). In some legs (17/72), the phenotype was milder and only a small number of neurons were misplaced (arrowhead in L). Milder phenotype was observed mostly in T1 legs (15/17). (M) The presence of a BAC transgene containing the *unc-4* locus restored the organization of the chordotonal neurons (36/36 legs, N = 6 animals) in the *unc-4* mutant background. In E-H, A:anterior; P:posterior; V:Ventral; D; Dorsal Whisker box plots in I: the box delineates the first and third quartiles, the line in the box is median, and whiskers showing minimum and maximum values within 1.5 times the interquartile range. Scale bar is 20 microns.

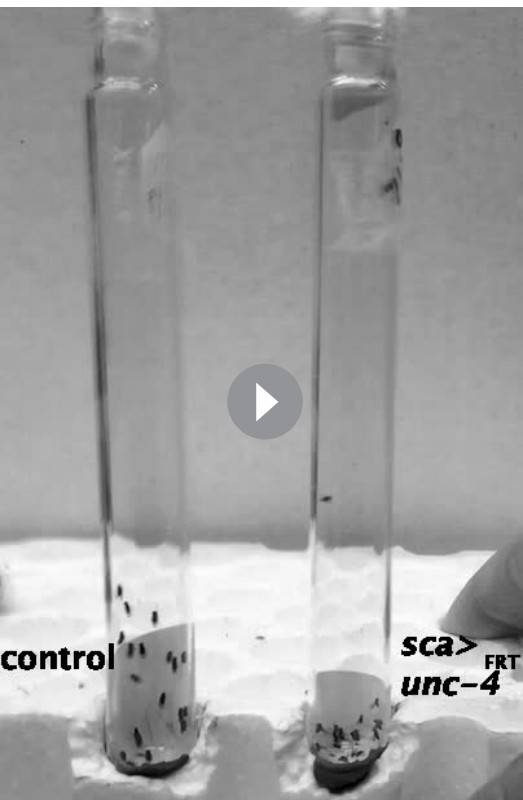

**Video 8.** Climbing activity of group of flies with the CNS and PNS specific deletion of *unc-4*: left, control (*unc-4*FRT /+; *sca*-GAL4/UAS-FLP.D); right, *sca >unc-4*FRT (*unc-4*FRT /y; *sca*-GAL4/UAS-FLP.D).

https://elifesciences.org/articles/55007#video8

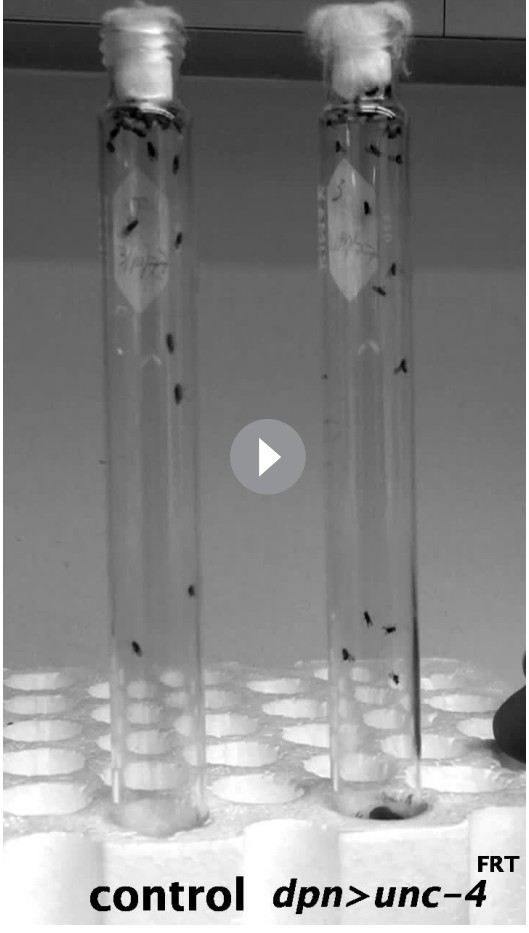

**Video 9.** Climbing activity of group of flies with the CNS specific deletion of *unc-4*: left, control (*unc-4*FRT /+; *dpn*EE-GAL4/UAS-FLP); right, *dpn*EE *>unc-4*FRT (*unc-4*FRT /y; *dpn*EE-GAL4/UAS-FLP).

https://elifesciences.org/articles/55007#video9

these motor neurons were the MN1-5 neurons, that innervate the indirect flight muscles, as well as motoneurons to neck and haltere muscles and the abdominal body wall (*Figure 4—figure supplement 1*). Most of these adult motoneurons come from remodeled larval motoneurons (*Consoulas et al., 2002*). Comparison of the adult cells with *unc-4*AD-*VGlut*DBD expression in the larval VNC indicated that these are remodeled versions of larval U/CQ motor neurons, from NB7-1 while others are likely embryonic-born neurons from NB2-4 lineage (*Figure 4—figure supplement 2*). In the VNC of *unc-4* mutant adults, these motor neurons were still present and marked with *unc-4*AD-*VGlut*DBD, but they were disorganized and reduced in size (*Figure 4C*; the MN5 size: 283+/- 54 µm$^2$ in controls vs 94+/- 31 µm$^2$ in mutants; N = 10 and 14 cells, respectively). Unc-4, thus, is expressed in embryonic-born remodeled adult motor neurons and functions to promote their growth.

Finally, we intersected *unc-4*DBD with *gad1*AD, a line that is expressed in all GABAergic neurons, to assess whether Unc-4 regulates GABA fate. In control animals, we failed to detect any GABAergic Unc-4+ neurons in the VNC (*Figure 4D*). RNA in situ hybridization against *gad1* and *ChAT* genes confirmed this and our previous published findings and showed that all Unc-4+ postembyonic hemilineages in the VNC express *ChAT* mRNA but not *gad1* mRNA (*Figure 4—figure supplement 2*). In *unc-4* mutants, however, *unc-4*DBD-*gad1*AD marked many neurons in the VNC (*Figure 4D*). Presence of a BAC clone containing *unc-4* locus reversed this expression pattern, significantly reducing the number of ectopic GABAergic neurons (*Figure 4E*). We identified these cells as a subset of the cholinergic hemilineage 11A (*Harris et al., 2015*; *Lacin et al., 2019*; *Shepherd et al., 2019*) based on the following observations: (i) like 11A neurons, they reside in posterior and dorsal region of only T1

**Table 1.** Three-leg rubbing and climbing behavior in response to tissue specific removal of Unc-4.

| Driver | Target cell types | Success in three-leg rubbing | | Success in climbing, time to climb[§] (in seconds) | |
|---|---|---|---|---|---|
| | | Control | unc-4 removal | Control | unc-4 removal |
| sca-GAL4 | All CNS and PNS neurons | 88.9%, N = 18 | 0%[*], N = 20 | 100%, 17.7 s. +/- 8, N = 22 | 9.1%[*], N/A[¶], N = 22 |
| dpn-GAL4 | All CNS neurons | 81.8%, N = 22 | 28.5%[*], N = 21 | 90%, 20.1 s. +/- 15, N = 30 | 80%[†], 27.6 s.[‡]+/- 20.5, N = 28 |
| ato-Gal4 | Chordotonal neurons | 90%, N = 33 | 14.8%[*], N = 27 | N/A[**] | N/A[**] |

[*]Significant change, Fisher exact test P value < 0.001.

[†]Non-significant change, Fisher exact test p value>0.1.

[‡]Non-significant change, Mann-Whitney U Test p value>0.1.

[§]failed climbing attempts not included for quantification.

[¶] not assessed since most animals failed to climb.

[**]ato-GAL4 >FLP expression impaired the negative geotaxis in both control and experimental animals, thus climbing assay was not performed.

and T2 segments (*Figure 4D*), (ii) their axonal trajectories match that of the 11A neurons (*Figure 4F–G*), (iii) they express Nkx6, a marker for 11A neurons (*Figure 4H*), and (iv) they intermingle with Eve[+] 11B neurons, which are the sibling neurons of 11A neurons (*Lacin and Truman, 2016*; *Lacin et al., 2019*; *Figure 4I*).

To test whether the transformed 11A neurons in the *unc-4* mutants are purely GABAergic or exhibit mixed GABAergic/cholinergic identity, we assayed them for the presence of *ChAT* and *gad1* mRNA (*Figure 4J*) and found that 92 +/- 5% expressed gad1 mRNA, 2 +/- 4% expressed *ChAT* mRNA, and none co-expressed both (N = 93 cells, 5 CNSs). Thus, most of these 11A neurons switch from a cholinergic to a GABAergic fate. Moreover, most of these converted neurons show GABA production as well as the *gad1* mRNA (*Figure 4K*). Unc-4, thus, appears to act in 11A hemilineage to promote the cholinergic fate and repress the GABAergic fate. Importantly, misexpression of Unc-4 in immature neurons GABAergic VNC lineages, such as 13A and 19A, throughout development did not, however, alter their GABAergic fate. Consequently, Unc-4 is not a general repressor of the GABAergic fate (N = 5 animals; not shown).

11A neurons likely function in take-off behavior of flies (*Harris et al., 2015*; *Kennedy and Broadie, 2018*). We could not investigate if the ectopic GABA fate observed in 11A neurons lead to any behavioral defect because we lacked a driver to remove Unc-4 only from this hemilineage.

## Unc-4 function is required for proper neuronal projections in 7b neurons

To determine if loss of Unc-4 function leads to defects in the anatomy of neurons in the Unc-4[+] hemilineages, we visualized them in both mutant and control flies. Unc-4 is expressed in hemilineages 7B, 11A, 12A, 17A, 18B,19B, and 23B. We used lineage-specific GAL4 lines built for this study (see methods) together with NB intersected reporter immortalization technique (*Awasaki et al., 2014*) to individually visualize the 7B, 12A, 18B, 23B hemilineages (*Figure 5A–D*). We also used the immortalization technique for the 11A and 19B neurons,

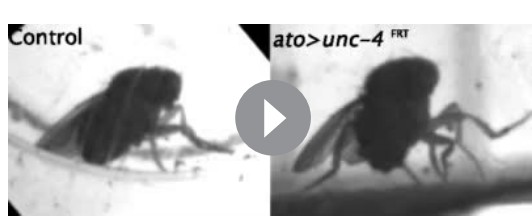

**Video 10.** Three-leg rubbing activity of individual flies with the chordotonal organ lineage specific deletion of *unc-4*: left, control (*unc-4*[FRT] /+; *ato*-GAL4[3.6]/UAS-FLP); right, *ato* >*unc-4*[FRT] (*unc-4*[FRT] /y; *ato*-GAL4[3.6]/UAS-FLP).
https://elifesciences.org/articles/55007#video10

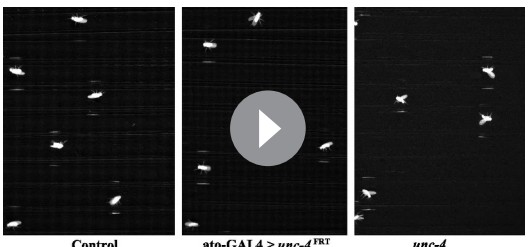

**Video 11.** Walking activity of individual flies in small-diameter tubes: left, control (*unc-4*[FRT] /+; *ato*-GAL4[3.6]/ UAS-FLP); middle, *ato* >*unc-4*[FRT] (*unc-4*[FRT] /y; *ato*-GAL4[3.6]/UAS-FLP); right, *unc-4* mutant (*unc-4*[DBD] /y).
https://elifesciences.org/articles/55007#video11

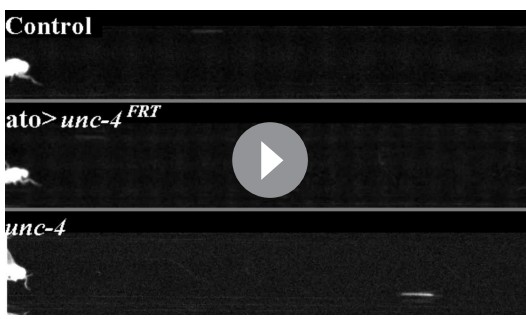

**Video 12.** Magnified version of Video 11. Single fly shown for each gentype.
https://elifesciences.org/articles/55007#video12

but the anatomy was complicated by the presence of their surviving sibling hemilineages (not shown). We could not evaluate 17A neurons due to lack of reagents. The loss of Unc-4 function had no gross defect on neuronal projections with the notable exception of hemilineage 7B. These neurons exhibited altered projections into the leg neuropils in the absence of Unc-4 function (*Figure 5A*; see below).

Lineage 7B neurons are present in all thoracic segments, but only neurons of the T2 segment innervate the leg neuropils (*Shepherd et al., 2019*). Using multicolor flip-out clones in wild-type animals (*Nern et al., 2015*), we found that many 7B neurons in T2 send ipsilateral projection to the T2 leg neuropil, others project into all of the contralateral leg neuropils, and others do not innervate the leg neuropil at all (*Figure 6A*; not shown). Of note, we found that the projections of 7B neurons into the ipsilateral leg neuropil have synaptic output sites, indicating that they communicate to other neurons to execute leg-related behaviors (*Figure 6B*). We used two different reporter immortalization strategies as well as MARCM approach to follow axonal projections of hemilineage 7B neuron in the absence of Unc-4 (see methods). All three gave the same result: loss of *unc-4* severely reduced 7B projections into the ipsilateral leg neuropils and also reduced contralateral leg projections (*Figure 6C–I*). Unc-4, therefore, functions cell autonomously in 7B neurons to support their proper innervation of the leg neuropil.

## Unc-4 promotes the development of flight take-off circuit

The anatomical changes in the 7B neurons with the loss of Unc-4 function is paralleled by a deficit in take-off behavior. Previous work showed that activation of 7B neurons elicits take-off behavior in decapitated flies (*Harris et al., 2015*). We used *ey*-AD;*dbx*DBD driven FLP expression in *unc4*FRT flies to remove Unc-4 specifically in 7B neurons (*Figure 7A,B*; *Figure 7—figure supplement 1*). To test the take-off behavior of the resulting flies, we first assessed their response to banging the vial in which they are housed (*Video 13*). Control animals responded to this stimulus by exhibiting a take-off behavior. In contrast, 7B specific Unc-4 knock-out flies raised their wings in an apparent attempt to take off, but they failed to be airborne. Some were able to take off after an unusual duration and multiple rounds of wing raise. We then recorded the take-off behavior of these flies with a high-speed camera after eliciting escape response via visual stimuli (*Williamson et al., 2018*). This analysis revealed that 7B specific Unc-4 knock-out flies process the visual stimulus, initiate the take-off behavior by adjusting their posture and raising their wings, but fail to extend their middle legs to cause lift off (*Figure 7B,C*; *Video 14*). Interestingly, some of these flies initiated the flight by just wing flapping without any clear jump (*Video 15*). These results demonstrate that hemilineage 7B specific removal of Unc-4 impairs take-off behavior.

The altered behavior seen in the flies that have lost Unc-4 function in their 7B neurons could result from the loss of function of a critical component in the take-off circuit or a mis-wiring of the cells to make novel connections that interfere with take-off. For example, in *C. elegans*, loss of Unc-4 function alters locomotion behavior due to ectopic synaptic connections of Unc-4+ neurons. We investigated whether the observed take-off phenotype was due to the gain of a novel function for the 7B neurons by ablating them during development via expressing the apoptotic gene, *head involution defective* (*hid*) in NB3-2 (*Bergmann et al., 1998*). The resulting animals behaved essentially identical as 7B specific *unc-4* mutants (*Video 16*), indicating the impairment take-off behavior is due to the loss of 7B neuronal function. Based on our findings, we conclude that Unc-4 acts in 7B neurons to promote their leg neuropil innervation, connections that are required for flies to use their legs properly during take-off.

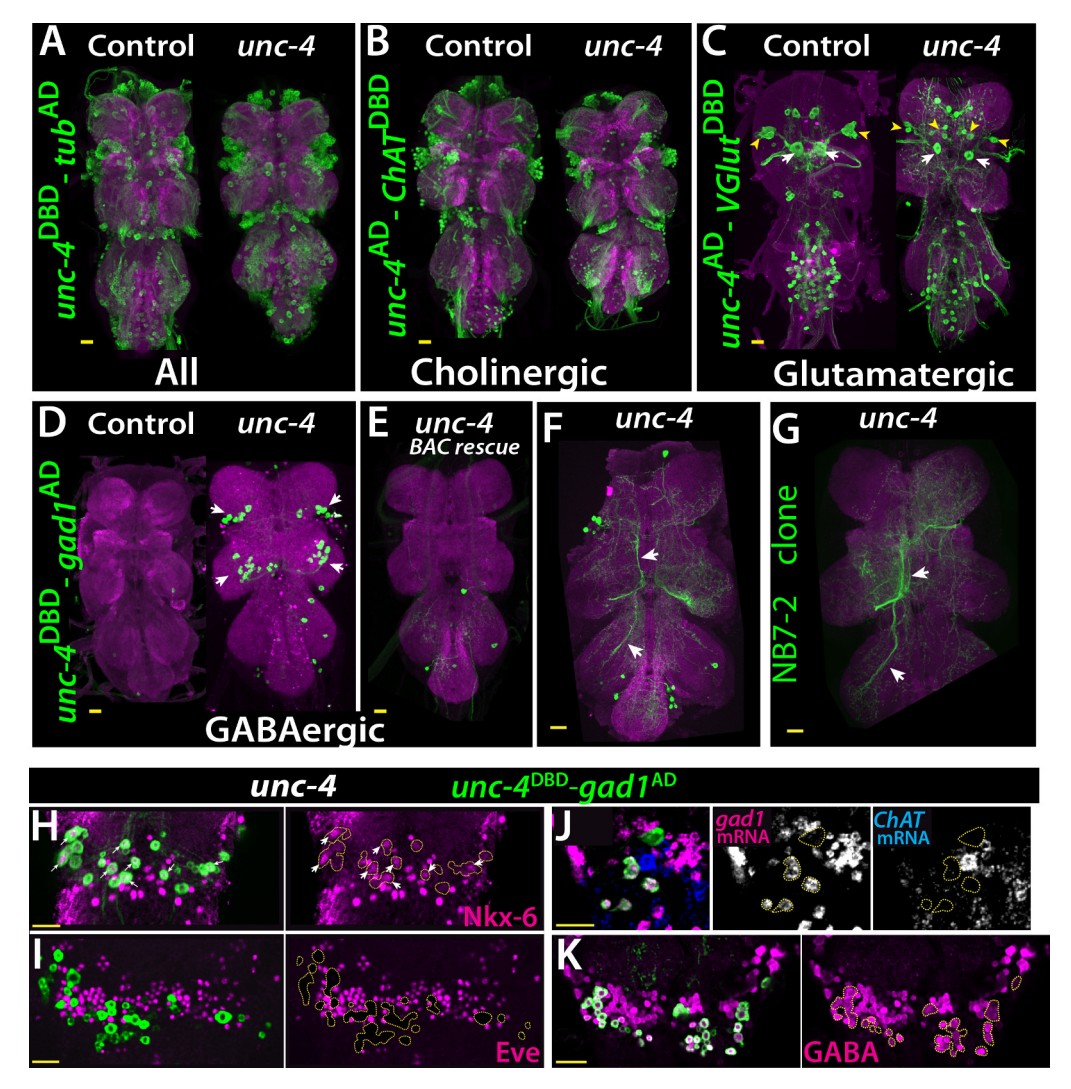

**Figure 4.** Cholinergic 11A neurons adopt GABAergic identity in *unc-4* mutant animals. (A–D) The adult VNC from control (left) and *unc-4* mutant (right) animals shown; images are from maximum Z projection of confocal slices. (A) All Unc-4+ neurons were visualized with *unc-4*DBD-*tub*AD driven GFP expression (green). No major difference was observed between the control and mutant. (B) *unc-4*AD-*ChAT*DBD split-GAL4 combination used to label cholinergic Unc-4+ neurons. Most Unc-4+ neurons were marked with this combination in both the control and mutant VNC and no major difference was seen between two. (C) *unc-4*AD-*VGlut*DBD driver marked a number of embryonic-born Unc-4+ glutamatergic motor neurons in the control VNC, which included flight related motor neurons MN1-4 (arrowheads) and MN5 (arrows). In the mutant, these motor neurons were disorganized and some showed reduction in cell size; for example, MN5 neurons. (D) *unc-4*DBD-*gad1*AD combination did not consistently show any GABAergic Unc-4+ neuron in the control CNS; however, in the *unc-4* mutant background GABAergic interneurons were found in both the T1 (9 +/- 5 neurons; N = 30) and T2 (17 +/- 12 neurons; N = 30) segments. (E) A BAC transgene containing *unc-4* locus suppressed the induced GABAergic phenotype and reduced the number of these neurons to 1.5 +/- 1 neurons per T1 segment and 1.3 +/- 1 neurons per T2 segment (N = 11 VNCs each). (F) A partial Z projection showing the main neuronal processes of *unc-4*DBD-*gad1*AD marked neurons in the VNC of an *unc-4* mutant animal. They extend ipsilateral processes both anteriorly and posteriorly (arrows). (G) NB7-2 intersected reporter immortalization used to label all lineage 11 neurons in *unc-4* mutant background and partial Z projection made to show only 11A neuronal processes but not that of 11B. The projections by the mutant cells match those of the 11A neurons (arrows). (H–K) *unc-4*DBD-*gad1*AD expression (outlined with dashed yellow lines) in the T2 segment from *unc-4* mutants shown. Nkx-6, a marker for 11A neurons, is expressed in a subset of *unc-4*DBD-*gad1*AD marked neurons (arrows in H) while Eve, a marker for 11B neurons, is not (I). (J) *ChAT* (blue) and *gad1*(magenta) mRNAs were visualized via RNA in situ hybridization. Majority of *unc-4*DBD-*gad1*AD marked neurons expressed *gad1* mRNA but not *ChAT* mRNA. (K) *unc-4*DBD-*gad1*AD marked neurons were immunopositive for GABA reactivity (magenta). Scale bar is 20 microns.

The online version of this article includes the following figure supplement(s) for figure 4:

**Figure supplement 1.** Emryonic-born Unc-4+ neurons become flight motor neurons in the adult VNC.

**Figure supplement 2.** *ChAT* (magenta) and *gad1*(blue) mRNAs were visualized via RNA in situ hybridization.

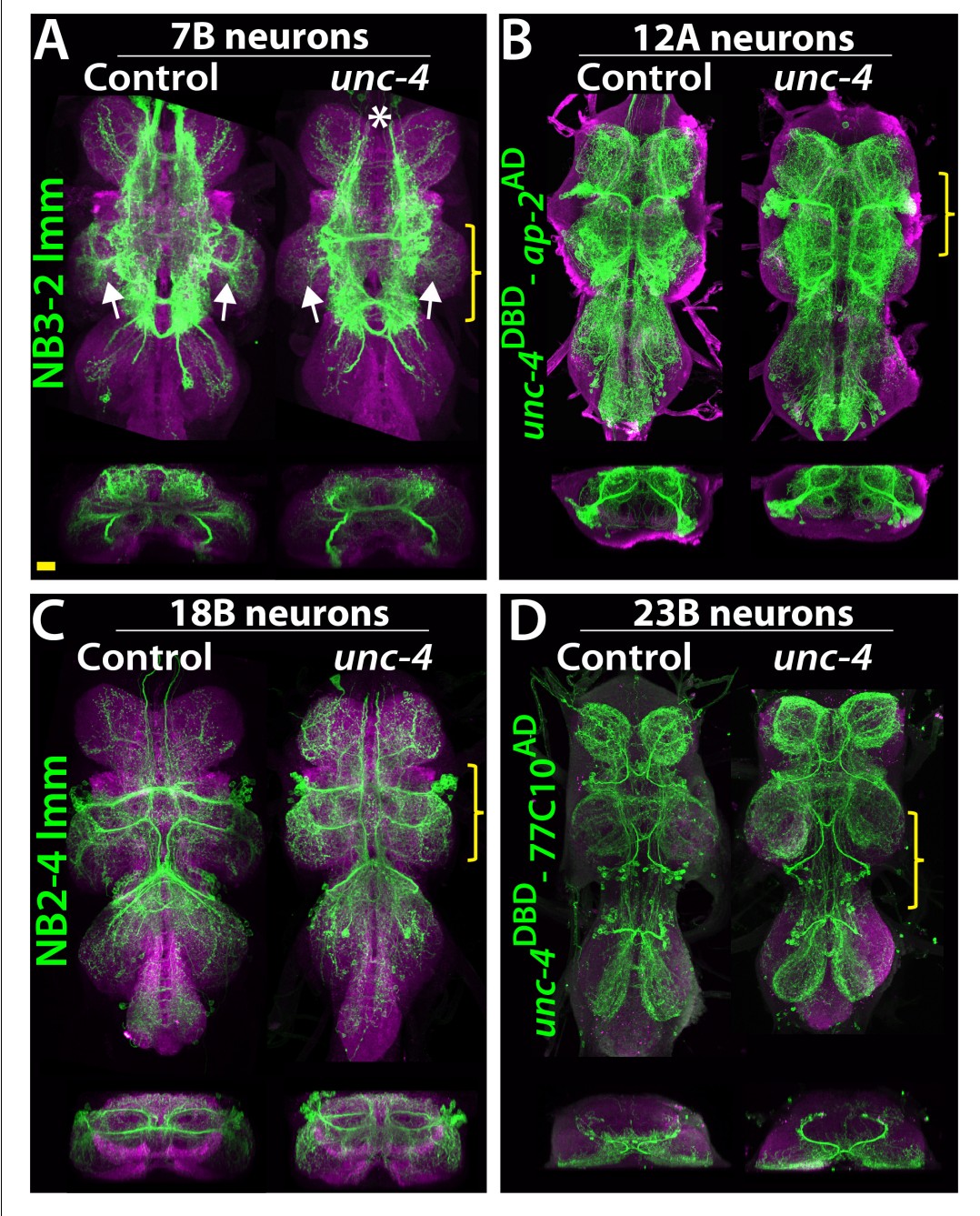

**Figure 5.** Projections of Unc-4+ lineages in control and *unc-4* mutant backgrounds. (**A–D**) VNC examples from control (left panels) and unc-4 mutant (right panels) adults. Transverse views from a 3D projection of the region indicated by yellow bracket are shown under the main images. (**A**) NB3-2 intersected reporter immortalization with *ey*-AD;*dbx*DBD driver was used to visualize the 7B neurons. In the unc-4 mutant animal, the 7B neurons did not innervate the T2 leg neuropil (compare the regions indicated by arrows). Note that 7B neurons in the mutant T1 segment were not labeled in this sample (asterisk), but in other mutant samples, projections of 7B neurons in T1 segments were similar to the control (not shown). (**B**) uncDBD- *AP-2*AD driver GFP used to mark 12A neurons. No dramatic difference was observed between the control and mutant VNCs. (**C**) Poxn-GAL4 driven reporter immortalization in NB2-4 used to visualize 18B neurons. Their neuronal projections appeared similar in the control and mutant. (**D**) *unc-4*DBD;77C10AD driven GFP expression used to visualize projections of 23B neurons, which did not differ between the control and mutant. Z projection images were made from a subset of confocal slices to visualize projections of all neurons. Scale bar is 20 microns.

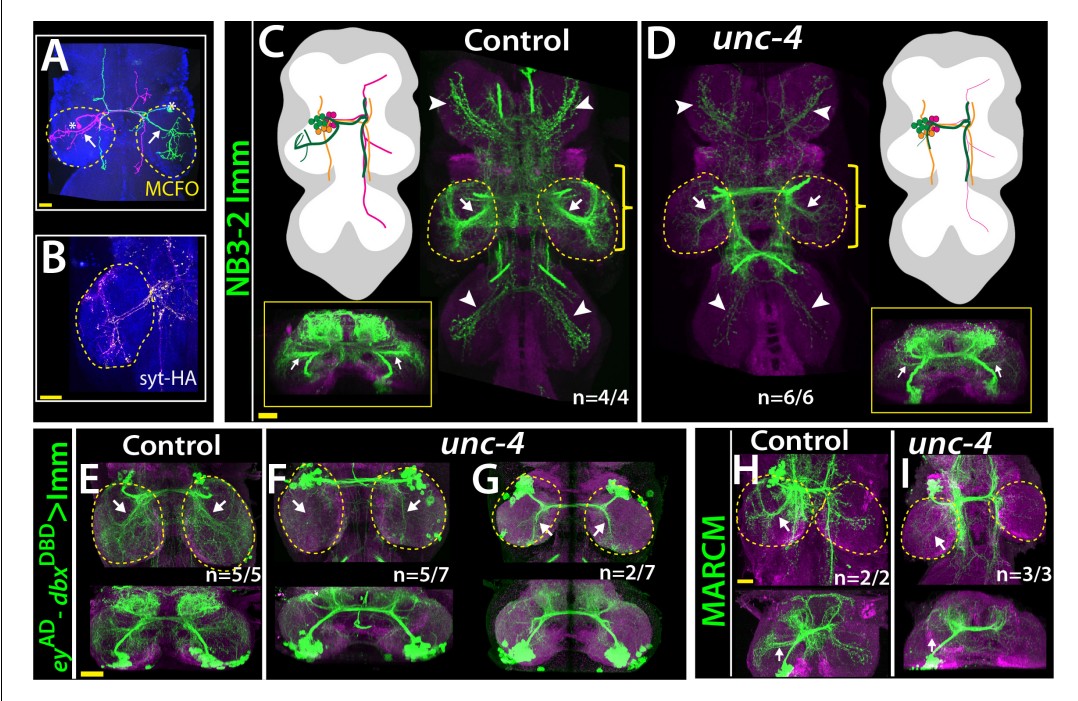

**Figure 6.** Unc-4 function is necessary in 7B neurons for the leg neuropil innervation . (**A**) An adult VNC containing MCFO clones showing two 7B neurons in the T2 segment; both extend projections (arrows) to the ipsilateral leg neuropil. (**B**) A subset of 7B neurons (magenta) in the T2 segment marked with SS20852-GAL4 driven *syt*-HA (yellow) showing synaptic outposts in the leg neuropil. (**C, D**) NB3-2 intersected reporter immortalization via *ey*-AD;*dbx*DBD driver used to visualize projections of 7B neurons in the control (**C**) and mutant (**D**) animals. Cartoons in the left panels schematized projections of the T2 7B neurons observed in confocal images. Transverse views (corresponding the yellow brackets) shown under the cartoons. In the mutant, ipsilateral (arrows) and contralateral (arrowheads) projections in the leg neuropils were reduced. Note that the numbers of T2 7B neurons marked with the reporter in control and mutant animals was the same (67 + - 6 and 68+/- 9 neurons in control and mutant animals, respectively; N = 7 clones for each). (**E–G**) A different immortalization technique involving FLP/FRT induced LexA:p65 without NB intersection used to mark 7B neurons in the VNC of control (**E**) and mutant (**F,G**) animals. *ey*-AD;*dbx*DBD expression was immortalized without NB intersection. Top panels show maximum z projections; bottom panels show the transverse view of the same region. As with the above manipulations, mutant 7B neurons failed to show the normal innervation of the leg neuropil (compare arrows in E and F). In some mutant animals, ipsilateral 7B projections extended posteriorly without innervating leg neuropils (arrows in G). (**H–I**) 7B MARCM clones in T2 segments of the control (**H**) and (**I**) *unc-4* mutant animals shown. Both ipsilateral (arrows) and contralateral (arrowheads) projections into leg neuropils were reduced in the mutant clone (**I**). Control clone has 68 cells; *unc-4* clone has 73 cells. (**C–H**) Maximum z projections were made of a subset of confocal slices to reveal neuronal processes in leg neuropils. n indicates the number of animals showing the phenotype. Leg neuropils are outlined with yellow dashed lines.

## Discussion

Using precise genetic tools, we dissected the function of the Unc-4 transcription factor in a lineage-specific manner. We found that within the PNS, Unc-4 function is needed for the proper development of the leg chordotonal organ and walking behavior; whereas in the CNS, Unc-4 dictates neurotransmitter usage within lineage 11A and regulates axonal projection and flight take-off behavior in lineage 7B. Below, we discuss three themes arising from our work: lineage-specific functions of individual transcription factors, an association of Unc4+ lineages with flight, and the lineage-based functional organization of the CNS in flies and vertebrates.

### Lineage -specific Unc-4 functions

Seven neuronal hemilineages express Unc-4 in the adult VNC, but our phenotypic studies revealed a function for Unc-4 in only two of them: in the 11A hemilineage, Unc-4 promotes the cholinergic fate and inhibits the GABAergic fate, while in the 7B hemilineage, Unc-4 ensures proper flight take-off behavior likely by promoting the proper projection patterns of the 7B interneurons into the leg neuropil. Why did we fail to detect a loss-of-function phenotype for Unc4 in most of the hemilineages in which it is expressed? A few reasons may explain this failure. First, our phenotypic analysis was

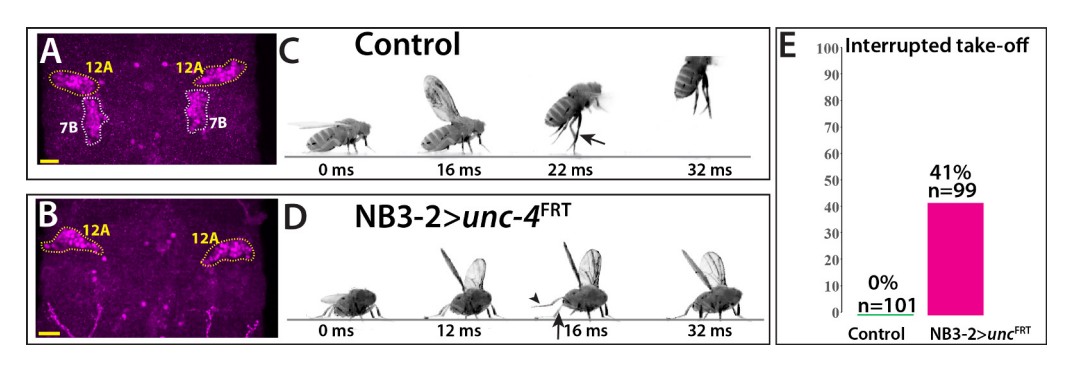

**Figure 7.** Removal of Unc-4 specifically in 7B neurons impairs take-off behavior. (A–B) Unc-4 expression (magenta) in the T1/T2 region of the VNC shown. (A) Unc-4 expression in T1 12A and T2 7B neurons are outlined with dashed lines. (B) NB3-2-GAL4 (ey$^{AD}$;dbx$^{DBD}$) driven FLP removes Unc-4 expression specifically from 7B neurons. (C–D) Sequences of still images shown from videos recording the response of flies to a looming stimulus. (C) The control animal exhibited a stereotyped take-off behavior. It first raised its wings, and then extended its T2 legs (arrow) to take off. (D) The NB3-2 specific Unc-4 knock-out animal exhibited an interrupted take-off behavior. It responded to the looming stimulus and raised its wings. It attempted to jump by lifting its hindleg (arrowhead) but did not extend its T2 legs (arrow), thus failed to take off. See also **Videos 14** and **15**. (E) The percentage of animals that showed interrupted take-off behavior shown. See Materials and methods section for more information. The time under each image is in milliseconds and indicates the elapsed time after the visual stimulus. Scale bar is 20 microns.

The online version of this article includes the following figure supplement(s) for figure 7:

**Figure supplement 1.** ey$^{AD}$;dbx$^{DBD}$ marks NB3-2 in the VNC.

limited: We tracked neuronal projection patterns and neurotransmitter fate, but not other molecular, cellular, or functional phenotypes. Unc-4 may function in other lineages to regulate other neuronal properties we did not assay, such as neurotransmitter receptor expression, channel composition, synaptic partner choice, and/or neuronal activity. In addition, as our analysis assayed all cells within the lineage, it would have missed defects that occur in single cells or small groups of cells within the entire hemilineage. Second, Unc-4 may act redundantly with other transcription factors to regulate the differentiation of distinct sets of neurons. Genetic redundancy among transcription factors regulating neuronal differentiation is commonly observed in the fly VNC (**Broihier et al., 2004**; **Lacin et al., 2009**). Thus, while our research clearly identifies a role for Unc-4 in two hemilineages, it does not exclude Unc-4 regulating more subtle cellular and molecular phenotypes in the other hemilineages in which it is expressed. Similarly, pan-neuronal deletion of Unc-4 specifically in the adult did not lead to any apparent behavioral defect even though Unc-4 expression is maintained in all Unc-4$^+$ lineages throughout adult life, suggesting

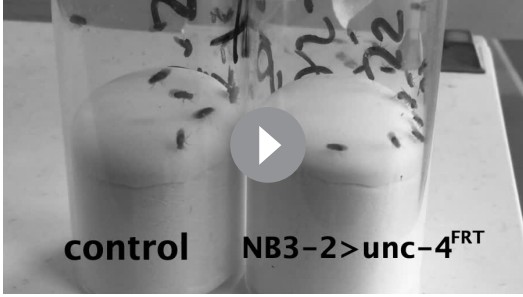

**Video 13.** The flight take-off behavior of flies with the 7B hemilineage specific deletion of unc-4 in response to banging the vial; 2X slower: left, control (5XUAS-FLP::PEST, unc-4$^{FRT}$/+; dbx$^{DBD}$/+; ey$^{AD}$/+); right, NB-3-2-GAL4 > unc-4$^{FRT}$ (5XUAS-FLP::PEST, unc-4$^{FRT}$/y; dbx$^{DBD}$/+; ey$^{AD}$/+).

https://elifesciences.org/articles/55007#video13

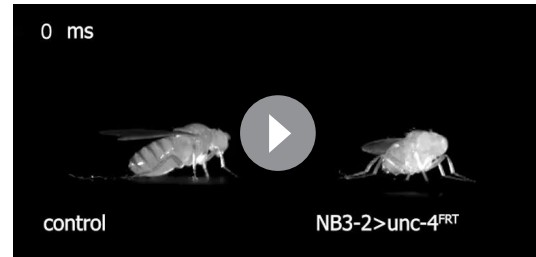

**Video 14.** The flight take-off behavior of flies with the 7B hemilineage specific deletion of unc-4 in response to visual looming stimuli; 200X slower: left, control (5XUAS-FLP::PEST, unc-4$^{FRT}$/+; dbx$^{DBD}$/+; ey$^{AD}$/+); right, NB-3-2-GAL4 > unc-4$^{FRT}$ (5XUAS-FLP::PEST, unc-4$^{FRT}$/y; dbx$^{DBD}$/+; ey$^{AD}$/+).

https://elifesciences.org/articles/55007#video14

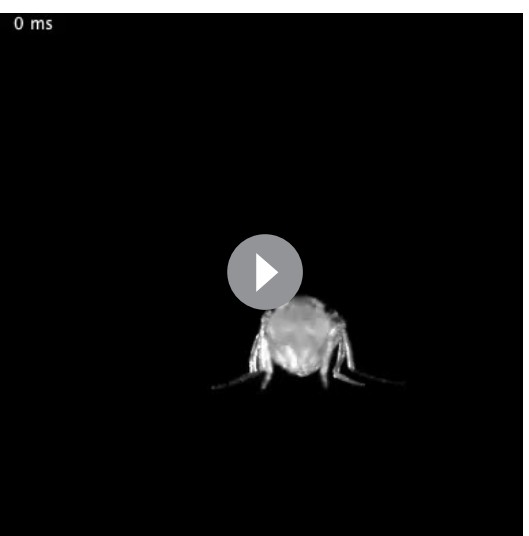

**Video 15.** Flies with 7B specific *unc-4* deletion initiating flight without take-off (5XUAS-FLP::PEST, *unc-4*FRT/y; *dbx*DBD/+; *ey*AD/+).
https://elifesciences.org/articles/55007#video15

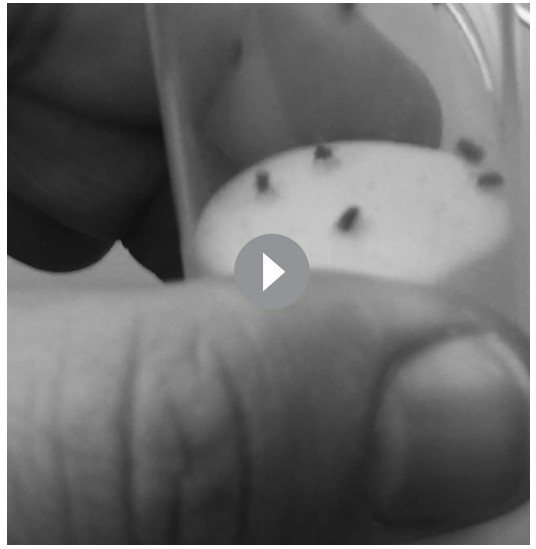

**Video 16.** The flight take-off behavior of flies with the ablated 7B hemilineage in response to banging the vial; 2X slower (UAS-hidAla5/+;; *dbx*DBD/+; *ey*AD/+).
https://elifesciences.org/articles/55007#video16

that Unc-4 function is dispensable in mature neurons after eclosion under standard lab conditions. Future work will be required to ascertain whether Unc-4 functions during adult life or in more than two of its expressing hemilineages during development. Nonetheless, our work shows that Unc-4 executes distinct functions in the 7B and 11A lineages. The Hox transcription factors, Ubx, Dfd, Scr, and Antp, have also been shown to execute distinct functions in different lineages in the fly CNS (*Marin et al., 2012*; *Kuert et al., 2014*), suggesting transcription factors may commonly drive distinct cellular outcomes in the context of different lineages. What underlies this ability of one transcription factor to regulate distinct cellular events in different neuronal lineages? The ancient nature of the lineage-specific mode of CNS development likely holds clues to this question. The CNS of all insects arises via the repeated divisions of a segmentally repeated array of neural stem cells whose number, ~30 pairs per hemisegment, has changed little over the course of insect evolution (*Thomas et al., 1984*; *Truman and Ball, 1998*). Within this pattern, each stem cell possessing a unique identity based on its position and time of formation. Each stem cell lineage has then evolved independently of the others since at least the last common ancestor of insects, approximately 500 million years ago. Thus, if during evolution an individual transcription factor became expressed in multiple neuronal lineages after this time, it would not be surprising that it would execute distinct functions in different neuronal lineages. The lineage-specific evolution of the CNS development in flies, worms, and vertebrates may explain why neurons of different lineages that share specific properties, for example, neurotransmitter expression, may employ distinct transcriptional programs to promote this trait (*Serrano-Saiz et al., 2013*; *Konstantinides et al., 2018*).

## Unc-4 and flight – a unifying theme?

Although Unc-4 appears to have distinct functions in different lineages, we found that an association with flight is a unifying feature among most Unc4+ interneuron lineages and motor neurons. All Unc-4+ hemilineages in the adult VNC except the 23B hemilineage heavily innervate the dorsal neuropils of the VNC, which are responsible for flight motor control and wing/haltere related behaviors, including wing-leg coordination (*Harris et al., 2015*; *Namiki et al., 2018*; *Shepherd et al., 2019*). For example, hemilineages 7B, 11A, and 18B regulate flight take-off behavior and 12A neurons control wing-based courtship singing (*Harris et al., 2015*; *Shirangi et al., 2016*). In addition, most Unc-4+ motor neurons are also involved with flight – these include MN1-5, which innervate the indirect flight muscles, as well as motor neurons that

innervate the haltere and neck muscles, which provide flight stabilization (*Figure 4C*). Since Unc-4 is conserved from worms to humans, it is likely that Ametabolous insects, like silverfish, which are primitively wingless, also have *unc-4*. It has yet to be determined, though, whether in such ametabolous insects the same hemilineages express Unc-4, and hence this pattern was in place prior to the evolution of flight. This would suggest that there was some underlying association amongst this set of hemilineages that may have been exploited in the evolution of flight. Alternatively, Unc-4 may be lacking in these hemilineages prior to the evolution of flight but then its expression may have been acquired by these hemilineages as they were co-opted into a unified set of wing-related behaviors.

## Lineage-based functional organization of the nervous system

The adult fly VNC is composed of 34 segmentally repeated hemilineages, which are groups of lineally related neurons with similar features for example, axonal projection and neurotransmitter expression (*Truman et al., 2004*; *Truman et al., 2010*; *Lacin et al., 2014*; *Harris et al., 2015*; *Lacin and Truman, 2016*; *Shepherd et al., 2016*; *Lacin et al., 2019*; *Shepherd et al., 2019*). These hemilineages also appear to function as modular units, each unit appears responsible for regulating particular behaviors, indicating the VNC is assembled via a lineage-based functional organization (*Harris et al., 2015*). The vertebrate spinal cord exhibits similar organization: lineally-related neurons acquire similar fates ('cardinal classes') and function in the same or parallel circuits (*Lu et al., 2015*). The similarity of the lineage-based organization in insect and vertebrate nerve/spinal cords raises the question whether they evolved from a common ground plan or are an example of convergent evolution. Molecular similarities in CNS development between flies and vertebrates support both CNS's arise from a common ground plan. For example, motor neuron identity in both flies and vertebrates, use the same set of transcription factors: Nkx6, Isl, and Lim3 (*Tsuchida et al., 1994*; *Pfaff et al., 1996*; *Thor and Thomas, 1997*; *Sander et al., 2000*; *Broihier and Skeath, 2002*; *Broihier et al., 2004*). Moreover, homologs of many transcription factors expressed in fly VNC interneurons, such as Eve and Lim1, also function in interneurons of the vertebrate spinal cord (*Lilly et al., 1999*; *Moran-Rivard et al., 2001*; *Pillai et al., 2007*; *Heckscher et al., 2015*). Whether any functional/molecular homology is present between fly and vertebrate neuronal classes awaits comparative genome-wide transcriptome analysis and functional characterization of neuronal classes in the insect VNC and vertebrate spinal cord.

# Materials and methods

## Key resources table

| Reagent type (species) or resource | Designation | Source or reference | Identifiers | Additional information |
|---|---|---|---|---|
| Antibody | Guinea pig anti-Sens polyclonal | PMID:10975525 | | 1:1000 dilution |
| Antibody | Rabbit anti-GABA polyclonal | Sigma | A2052 | 1:1000 dilution |
| Antibody | Chicken anti-GFP polyclonal | Life Tech. | A10262 | 1:1000 dilution |
| Antibody | Rabbit anti-HA polyclonal | Cell sig. | 3724S | 1:500 dilution |
| Antibody | Rabbit anti-GFP polyclonal | Thermo Fisher S. | A11122 | 1:1000 dilution |
| Antibody | Rabbit anti-Unc-4 polyclonal | PMID:24550109 | | 1:1000 dilution |
| Antibody | Rabbit Eve | PMID:2884106 | | 1:2000 dilution |
| Antibody | Mouse anti-Acj6 monoclonal | DSHB | Acj6 | 1:100 dilution |

*Continued on next page*

*Continued*

| Reagent type (species) or resource | Designation | Source or reference | Identifiers | Additional information |
|---|---|---|---|---|
| Antibody | Rat anti-Cadherin monoclonal | DSHB | DN-Ex#8 | 1:25 dilution |
| Antibody | goat anti-rabbit Alexa Fluor 488 | Life Technologies | A-11034 | 1:500 dilution |
| Antibody | goat anti-rabbit Alexa Fluor 568 | Life Technologies | A-11011 | 1:500 dilution |
| Antibody | goat anti-rabbit Alexa Fluor 633 | Life Technologies | A-21070 | 1:500 dilution |
| Antibody | goat anti-chicken Alexa Fluor 488 | Life Technologies | A-11039 | 1:500 dilution |
| Antibody | goat anti-mouse Alexa Fluor 568 | Life Technologies | A-11004 | 1:500 dilution |
| Antibody | goat anti-mouse Alexa Fluor 633 | Life Technologies | A-21050 | 1:500 dilution |
| Antibody | goat anti-rat Alexa Fluor568 | Life Technologies | A-11077 | 1:500 dilution |
| Antibody | goat anti-rat Alexa Fluor633 | Life Technologies | A-21094 | 1:500 dilution |
| Genetic reagent (*D. melanogaster*) | MKRS, P{ry[+t7.2]=hsFLP}86E/TM6B, P{w[+mC]=Crew}DH2, Tb[1] | Bloomington *Drosophila* Stock Center | RRID:BDSC_1501 | |
| Genetic reagent (*D. melanogaster*) | P{ry[+t7.2]=hsFLP}1, w[1118]; Adv[1]/CyO | Bloomington *Drosophila* Stock Center | RRID:BDSC_6 | |
| Genetic reagent (*D. melanogaster*) | y1 w*; P{UAS-FLP.D}JD1 | Bloomington *Drosophila* Stock Center | RRID:BDSC_4539 | |
| Genetic reagent (*D. melanogaster*) | w*; P{UAS-p35.H}BH1 | Bloomington *Drosophila* Stock Center | RRID:BDSC_5072 | |
| Genetic reagent (*D. melanogaster*) | P{Tub-dVP16AD.D} | Bloomington *Drosophila* Stock Center | RRID:BDSC_60295 | |
| Genetic reagent (*D. melanogaster*) | P{Tub-GAL4DBD.D} | Bloomington *Drosophila* Stock Center | RRID:BDSC_60298 | |
| Genetic reagent (*D. melanogaster*) | w[*]; P{w[+mC]=iav-GAL4.K}3 | Bloomington *Drosophila* Stock Center | RRID:BDSC_52273 | |
| Genetic reagent (*D. melanogaster*) | y[1] w[*]; Mi{y[+mDint2]=MIC}TfAP-2[MI04611]/TM3, Sb[1] Ser[1] | Bloomington *Drosophila* Stock Center | RRID:BDSC_37965 | |
| Genetic reagent (*D. melanogaster*) | 13XLexAop2-IVS-myr::GFP in attP40 | Bloomington *Drosophila* Stock Center | RRID:BDSC_32210 | |
| Genetic reagent (*D. melanogaster*) | w[*]; P{w[+mC]=tubP-GAL80[ts]}2/TM2 | Bloomington *Drosophila*Stock Center | RRID:BDSC_7017 | |
| Genetic reagent (*D. melanogaster*) | hsFlp2::PEST;; HA_V5_FLAG-MCFO | PMID:25964354 | RRID:BDSC_9494 | |

*Continued on next page*

*Continued*

| Reagent type (species) or resource | Designation | Source or reference | Identifiers | Additional information |
|---|---|---|---|---|
| Genetic reagent (*D. melanogaster*) | y1 w*; P{ato-GAL4.3.6}10 | PMID:10774724 | | |
| Genetic reagent (*D. melanogaster*) | Dpn-EE-GAL4-attp16;T3/T6b | PMID:26700685 | | |
| Genetic reagent (*D. melanogaster*) | pJFRC29-10XUAS-IVS-myr::GFP-p10 in attP40 | PMID:22493255 | | |
| Genetic reagent (*D. melanogaster*) | pJFRC105-10XUAS-IVS-nlstdTomatoin VK00040 | PMID:22493255 | | |
| Genetic reagent (*D. melanogaster*) | Poxn-GAL4.13 | PMID:12421707 | | |
| Genetic reagent (*D. melanogaster*) | y[1] w[*]; Mi{y[+mDint2]=MIC}TfAP-2[MI04611]/TM3, Sb[1] Ser[1] | Bloomington *Drosophila*Stock Center | RRID:BDSC_37965 | |
| Genetic reagent (*D. melanogaster*) | UAS-hidAla5 | PMID:9814704 | | |
| Genetic reagent (*D. melanogaster*) | Actin5Cp4.6>dsFRT > nlsLexAp65 | G Rubin | | |
| Genetic reagent (*D. melanogaster*) | pJFRC51-3XUAS-IVS-Syt::smGFP-HA in attP40 | G Rubin | | |
| Genetic reagent (*D. melanogaster*) | pJFRC12-10XUAS-IVS-myr::GFP in attP18 | G Rubin | | |
| Genetic reagent (*D. melanogaster*) | *unc-4*-GAL4 | this study | FBgn0024184 | built by H. Lacin |
| Genetic reagent (*D. melanogaster*) | *unc-4*-AD | this study | FBgn0024184 | built by H. Lacin |
| Genetic reagent (*D. melanogaster*) | *unc-4*-DBD | this study | FBgn0024184 | built by H. Lacin |
| Genetic reagent (*D. melanogaster*) | *unc-4*-FRT | this study | FBgn0024184 | built by H. Lacin |
| Genetic reagent (*D. melanogaster*) | *unc-4*-null | this study | FBgn0024184 | built by H. Lacin |
| Genetic reagent (*D. melanogaster*) | *TfAP-2*-AD | this study | FBgn0261953 | built by H. Lacin |
| Genetic reagent (*D. melanogaster*) | *TfAP-2*-GAL4 | this study | FBgn0261953 | built by H. Lacin |
| Genetic reagent (*D. melanogaster*) | w*;sca-GAL4 | PMID:2125959 | | |
| Chemical compound, drug | Paraformaldehyde | EMS | 15713 | |
| Chemical compound, drug | Vectashield | Vector | H-1000 | |
| Chemical compound, drug | Prolong Diamond | Molecular Probes | P36961 | |
| Recombinant DNA reagent | pHD-DsRed-attP | Addgene | 51019 | |
| Recombinant DNA reagent | pCFD4-U6:1_U6:3tandemgRNAs | Addgene | 49411 | |

*Continued on next page*

*Continued*

| Reagent type (species) or resource | Designation | Source or reference | Identifiers | Additional information |
|---|---|---|---|---|
| Recombinant DNA reagent | pBS-KS-attB2-SA(1)-T2A-Gal4-Hsp70 | Addgene | 62897 | |
| Recombinant DNA reagent | pBS-KS-attB2-SA(1)-T2A-Gal4DBD-Hsp70 | Addgene | 62903 | |
| Recombinant DNA reagent | pBS-KS-attB2-SA(1)-T2A-p65AD-Hsp70 | Addgene | 62914 | |
| Recombinant DNA reagent | pBS-KS-attB2-SA(0)-T2A-Gal4-Hsp70 | Addgene | 62896 | |
| Recombinant DNA reagent | pBS-KS-attB2-SA(0)-T2A-Gal4DBD-Hsp70 | Addgene | 62902 | |
| Recombinant DNA reagent | pBS-KS-attB2-SA(0)-T2A-p65AD-Hsp70 | Addgene | 62912 | |

## Transgenic animals unc-4$^{FRT}$ line

Via CRISPR and homology directed repair, we inserted two FRT sites at the same orientation into the first and third *unc-4* introns so that FRT sites flank the second and third exons, which encode the homeodomain.

### Guide RNAs

We used two guide RNAs to direct Cas9 activity to the first and third introns of *unc-4*. pCFD4 vector was used to generate tandem guide RNA expression platform with the following primers (*Port et al., 2014*). Forward primer: TATATAGGAAAGATATCCGGGTGAACTTCGG ttttcggggtcccatgtgtgGTTTTAGAGCTAGAAATAGCAAG. Reverse primer: ATTTTAACTT GCTATTTC TAGCTCTAAAACaagtctcaactaccatcgaCGACGTTAAATTGAAAATAGGTC Underlined region represents sequence from *unc*-4 locus.

### Donor DNA

Our donor construct contained three major elements: (i) the left and (ii) right homology arms, which are immediately adjacent to Cas9 cutting sites, and between them, (iii) the replacement arm which contained two FRTsites, an attp site, 3xP3-dsRED mini gene and the intervening unc-4 sequence to keep *unc-4* exons intact. *unc-4* genomic regions were cloned from a BAC clone (BACR01N10). PCR amplified DNA blocks were cloned into pHD-DsRed-attP vector via AarI and SapI enyzme sides sequentially as previously described (*Gratz et al., 2014*).

### Block 1

Primes used to generate the block 1, which contained the left homology arm (~1.6 kb) and the FRT sequence (underlined).

Forward primer (with AarI site): AGCCACACCTGCGAATTCGCTTCCAGTTGTCGGGCAC TCCAAAT.

Reverse primer (with AarI and FRT sites):CCAACACCTGCAAACCTACGAAGTTCCTATACTTTC TAGAGAATAGGAACTTCGAAAATTACCCAAAAATGGAAAACGC. The amplified DNA was inserted to the pHD-DsRed-attP vector via AarI enzyme site.

### Block 2

Two PCR products (blocks 2A and 2B) were ligated to generate Block 2. Primes used to generate the block 2A, which contained 2$^{nd}$ and 3$^{rd}$ *unc-4* exons and some intronic sequence. Forward primer (with SapI and XbaI sites): TTTCTCTAGATGGGGCTCTTCCTATGTCCCATGTGTGTGGAGTTCGGT. Reverse primer (with BamH1 site): ATTCGGGATCCAGTTGAGACTTTGGTATAGCTTTTAAATTGC . Primes used to generate the block 2B, which contained the right homology arm (~1.6 kb) and the FRT sequence (underlined). Forward primer (with BamH1): GGTGGGATCCGAAGTTCCTATTCTC

TAGAAAGTATAGGAACTTCCCCATCGAATATCTGATTATTAACAC. Reverse primer (with SapI and HindIII sites): TTCACAAAGCTTCACACTGCTCTTCAGACTGTCCTTAGTCAATAGACTTCTATTAC.

Block2A and Block2B were conjugated via BamH1site and inserted into a pUC19 vector via HindIII and XbaI sites in the same ligation reaction. Block 2 was recovered with SapI digestion and inserted into Sap1 site of the previously built construct, Block 1 in pHD-DsRed-attP.

The content of the donor and guide RNA vectors was verified by Sanger sequencing. A mixture of these vectors was injected into vas-Cas9 flies (w[1118]; PBac(y[+mDint2]=vas-Cas9)VK00027) by Rainbow transgenics (https://www.rainbowgene.com).

The transformed animals were identified based on dsRED expressions and stable stocks were generated. PCR analysis confirmed the expected editing. unc-4 Reporter lines: pBS-KS-attB2-SA(1)-T2A-Gal4-Hsp70, pBS-KS-attB2-SA(1)-T2A-Gal4DBD-Hsp70, pBS-KS-attB2-SA(1)-T2A-p65AD-Hsp70 vectors were microinjected into Unc-4$^{FRT}$ flies to generate unc-4$^{GAL4}$, unc-4$^{DBD}$, unc-4$^{AD}$ reporter lines, respectively via PhiC31-attP mediated integration. If necessary, LoxP flanked 3XP3-dsRED mini gene was removed from the reporter lines via Cre recombinase. Note that all the reporter lines behave as null mutant due to the stop codon introduced by Trojan constructs. Reporter lines were balanced with the FM7c balancer.

## unc-4$^{null}$ line

Second and third unc-4 exons were removed in the germline to generate unc-4$^{null}$ line. To do that, we crossed unc-4$^{FRT}$ line with hsFLP1 flies, and resultant larval progeny were heat shocked at 37°C for an hour. Adult female progeny was later crossed individually to FM7c males to isolate and generate unc-4$^{null}$ lines.

Note that the unc-4 locus over FM7 balancers showed a low rate of recombination/chromosomal rearrangement. When unc-4 mutant animals over FM7 balancer were outcrossed to males carrying a wild type X chromosome, we observed occasional non FM7 male flies, which were not unc-4 mutants. These flies lacked 3XP3-dsRED expression, which marks edited unc-4 locus and behaved like wild-type. Wild type unc-4 locus in the FM7 chromosome is located close to the Bar locus which is known to exhibit rare chromosomal arrangements (*Miller et al., 2016*).

## Lineage-specific GAL lines

AP-2$^{AD}$ and AP-2$^{GAL4}$ lines were generated from the TfAP-2$^{MI04611}$ MIMIC line via Trojan exon insertion as described before (*Diao et al., 2015*; *Nagarkar-Jaiswal et al., 2015*; *Lacin et al., 2019*). Hemilineage 12A specific expression of AP-2 was previously documented (*Etheredge, 2017*). unc-4$^{DBD}$;AP-2$^{AD}$ split-GAL4 combination was generated to mark specifically 12A neurons. ey-$^{AD}$;dbx$^{DBD}$ driver was built based on the information that Ey and Dbx are co-expressed only in NB3-2 in the VNC (*Lacin and Truman, 2016*; *Lacin et al., 2019*). 23B-specific driver unc-4$^{DBD}$;77C10$^{AD}$ was built based on the information that Unc-4 and Acj-6 are co-expressed only in 23B neurons and 77C10 reports Acj-6 expression.

## Cell labeling chemistry

### Immunochemistry

Samples were dissected in phosphate buffered saline (PBS) and fixed with 2% paraformaldehyde in PBS for an hour at room temperature and then washed several times in PBS-TX (PBS with 1% Triton-X100) for a total 20 min. Tissues were incubated with primary antibodies (Key Resources Table) for two to four hours at room temperature or overnight 4°C. After three to four rinses with PBS-TX to remove the primary antisera, tissues were washed with PBS-TX for an hour. After wash, tissues secondary antibodies were applied for two hours at room temperature or overnight at 4°C. Tissues were washed again with PBS-TX for an hour and mounted in Vectashield or in DPX after dehydration through an ethanol series and clearing in xylene (*Truman et al., 2004*).

### RNA in situ hybridization

Hybridization chain reaction-based RNA in situ hybridization was performed with commercially purchased probes as described before (*Lacin et al., 2019*).

## Lineage clone generation

NB intersected reporter immortalization clones (*Awasaki et al., 2014*; *Lacin and Truman, 2016*) in control and *unc-4* mutant background were generated by crossing NB specific GAL4 lines to the following lines respectively: [dpn >KDRT-stop-KDRT>Cre:PEST; 13XLexAop2-IVS-myr::GFP, Act5c > loxP stop loxP >LexA::p65; UAS-KD] and [dpn >KDRT-stop-KDRT>Cre:PEST, *unc-4*[null]; 13XLexAop2-IVS-myr::GFP, Act5c > loxP stop loxP >LexA::p65,; UAS-KD].

Reporter immortalization clones without NB intersection (7B clones in *Figure 6E–G*) were generated by using the following transgenic lines: [*ey*-[AD];*dbx*[DBD]], [*unc-4*[null] /FM7i, actGFP], [13XLexAop2-IVS-myr::GFP, Actin5Cp4.6>dsFRT > nlsLexAp65], and [UAS-FLP.D].

Multi-Color Flipout (MCFO) clones were stochastically generated from the progeny of the cross of SS20852-GAL4 (R34H12[AD]; VT040572[DBD]) line with R57C10Flp2::PEST; ; HA_V5_FLAG_OLLAS/MCFO-7 (*Nern et al., 2015*).

The MARCM technique was used to generate NB3-2 lineage clones (*Lee and Luo, 2001*). Newly hatched first instar larvae were incubated at 37°C for an hour and adult nerve cords were dissected, fixed and stained. The following lines were used: [hsFLPop, Tub-GAL80, FRT19A; act >FRT-stop-FRT>GAL4, UAS-CD8GFP], [*unc-4*[null] FRT19A/FM7i, actGFP], [FRT19A], [elavC155-GAL4].

## TARGET mediated Unc-4 removal

To remove Unc-4 globally starting from the early embryonic stages, embryos from the cross of 5XUAS-FLP::PEST, *unc-4*[FRT] line with *tubP*-GAL4; *tubP*-GAL80[ts] line (*McGuire et al., 2004*) were collected for 6 hr at 18°C and then transferred to and kept in 29°C until adults eclosed. 5–10 days old adults were used for behavioral analysis.

To remove Unc-4 globally starting from the late pupal stages, collected embryos from the same cross were kept at 18°C until animals reached the pharate adult stage and then transferred to and kept in 29°C until emerged adults aged for two weeks for behavioral analysis.

## Fly stocks and behavioral experiments

Flies were reared on the standard cornmeal fly food at 25°C unless indicated otherwise. Fly lines used in this study are listed in Key Resources Table. Fly behaviors were tested at room temperature (22–25°C). Flies were used 2–10 days post-eclosion. Flies with damaged legs and wings were excluded from the tests. Animal behaviors were recorded with a USB or iPad Pro camera except for stimuli induced take-off behaviors for which we used high speed cameras. Please see *Supplementary file 1* for genotypes of the animals used in each figure.

### Three leg grooming

Flies were individually observed under the microscope for three leg-rubbing behavior that involves either pairs of front or hind leg and one mid leg. Three episodes were quantified for each fly; success or failure was called based on the majority. If two initial episodes result in failure or success, flies were not tested for the third episode.

### Climbing

Flies were put individually into a 5 ml serological pipette with a 5 mm diameter and the pipette was banged and hold vertically to induce climbing. The event on which a fly climbed 14 cm in 1 min or under was called a success and the climbing time was documented. Events with the climbing time over 1 min was considered as a failure.

### Walking

Flies were put individually into sleep assay vials with a 4 mm dimeter and ends of the horizontally placedvials were plugged with cottons to give approximately 6 cm length for flies to walk. The locomotion of the flies was recorded with a USB camera. Walking speed were measured from the recorded locomotion tracts that were continuous i.e., not interrupted by fall or grooming via wrMTrck pluggin on ImageJ (*Husson et al., 2013*; *Brooks et al., 2016*).

ImageJ: http://imagej.nih.gov/ij/ and wrMTrck: http://www.phage.dk/plugins/wrmtrck.html.

## Flight Take-off

Flies individually were put on top of a large white surface (top of a chest freezer) and observed with naked eyes whether they take-off to fly away. The flies that were able to fly away was scored as successful.

## Visual stimuli induced take-off

Two kinds of visual stimuli were used to induce jumping in flies as described earlier (*Williamson et al., 2018*): (i) the looming stimulus with an azimuth/elevation of 90°/45° and (ii) the flickering stimulus, where the entire visual background scene alternated white and black at 4 Hz. Videos of the flies were recorded during the visual stimulus at a rate of 6000 frames per second. Normally, flies exhibit a stereotyped takeoff sequence: adjust posture, raise wings, extend mid-thoracic legs, initiate flight. For the quantification of the observed phenotype reported in this paper, flies that initiated the flight sequence without completing the takeoff sequence were scored as 'interrupted take-off.' The flickering stimulus was represented to 54 control and 37 NB3−2 > unc-4$^{FRT}$ flies; 0 and 16 flies showed take-off interruption, respectively. The looming stimulus was represented to 47 control and 60 NB3−2 > unc-4$^{FRT}$ flies; 0 and 24 flies showed take-off interruption, respectively. Quantification in *Figure 7E* was based on the results obtained from both stimuli.

## Tethered-flight assay

Insect pins were glued to the thorax dorsally and brief air puffs were blown to the flies to induce wing flapping. Flies that showed high frequency wing flapping were considered successful.

## Acknowledgements

We thank H Bellen, B Dickson, C Doe, and G Rubin for sharing reagents. We thank C Yang and T Lee for sharing their fly stocks. We are indebted to T Laverty, K Hibbard, A Cavallaro, Y Zhu and B Wilson for fly husbandry, and Alison Howard for administrative support. We thank D Shepherd, J Garcia-Marques, D Miller, J Ache for their advices in this study and thank D Shepherd again for sharing his unpublished data. This research is supported by HHMI to JWT and NIH NS083086 to JBS.

## Additional information

### Funding

| Funder | Grant reference number | Author |
|---|---|---|
| NIH Office of the Director | NS083086 | James B Skeath |
| Howard Hughes Medical Institute | | James W Truman |

The funders had no role in study design, data collection and interpretation, or the decision to submit the work for publication.

### Author contributions

Haluk Lacin, Conceptualization, Data curation, Formal analysis, Investigation, Visualization, Methodology, Writing - original draft, Writing - review and editing; W Ryan Williamson, Resources, Visualization, Methodology; Gwyneth M Card, Resources, Funding acquisition, Methodology; James B Skeath, Resources, Funding acquisition, Writing - review and editing; James W Truman, Supervision, Funding acquisition, Visualization, Project administration, Writing - review and editing

### Author ORCIDs

Haluk Lacin https://orcid.org/0000-0003-2468-9618
Gwyneth M Card http://orcid.org/0000-0002-7679-3639
James W Truman http://orcid.org/0000-0002-9209-5435

Decision letter and Author response
Decision letter https://doi.org/10.7554/eLife.55007.sa1
Author response https://doi.org/10.7554/eLife.55007.sa2

## Additional files

### Supplementary files

- Supplementary file 1. Genotypes of fly lines used in each experiment.

- Transparent reporting form

### Data availability

All data generated or analysed during this study are included in the manuscript and supporting files.

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
