## [Decision Letter]

**Acceptance summary:**

This study by Lacin and colleagues is a fine example of leveraging genetic reagents to incisively study the role of a gene (in this case the transcription factor, Unc-4) in the development and function of neuronal lineages. The work is robust and beautifully illustrated, focused on two hemi-lineages. There is lots to like about this study, particularly the clear logic and rigorous application of genetic techniques designed to squeeze out as much specificity as possible. The authors provided a clear, and perhaps technically the, as yet, most rigorously demonstrated example. This is not a new concept, so the question is around novelty. Yet, this type of work needs to be done and appreciated. Its rigour is impressive and it makes it clear that to understand how combinations of transcription factors regulate cellular properties, which then combine to shape specific behavioural outputs, remains a complex and challenging task.

**Decision letter after peer review:**

Thank you for submitting your work entitled "Unc-4 acts to promote neuronal identity and development of the take-off circuit in the *Drosophila* CNS" for consideration by *eLife*. Your revised article has been evaluated by K VijayRaghavan as the Senior Editor, a Reviewing Editor, and two reviewers.

A consolidated review is attached. The authors will see that they need to do next to nothing before acceptance, except read the words of praise and highlight better a significant finding and deal with the minor comments.

Summary and consolidated reviews:

This report uses the powerful genetic tools of *Drosophila* to characterize the expression and function of the conserved Unc-4 transcription factor during the development of the adult CNS. They show that Unc-4 is expressed in about half of the hemilineages that generate the adult ventral nerve cord. They use hemilineage specific Gal4 lines to determine the lineage-specific function of Unc-4, finding lineage-specific LOF phenotypes in two hemilineages. In the 11A hemilineage Unc-4 is required for promoting cholinergic identity and suppressing GABA identity, although Unc-4 is not capable of repressing GABA identity when ectopically expressed in other hemilineages. Thus, this is a lineage-specific phenotype. Similarly, Unc-4 LOF in hemilineage 7B results in failure to innervate T2 leg neuropil and a highly specific flight take-off defect (no jump).

Overall, the experiments are beautifully documented; the text is clearly written, and the conclusions are well supported. The positives of this paper are the importance of characterizing a highly conserved transcription factor, the quality of the experiments and conclusions, and the identification of an entry point for subsequent characterization of an adult flight "jump" neural circuit. The only negative, perhaps, is the lack of a single coherent story – which is the direct consequence of having lineage-specific phenotypes, and thus hard to avoid.

There is lots to like about this study, particularly the clear logic and rigorous application of genetic techniques designed to squeeze out as much specificity as possible. The results are clear: that in different hemilineages Unc-4 regulates different aspects of development, cell specification, and function. While here the authors provided a clear, and perhaps technically the as yet most rigorously demonstrated example, this is not a new concept. So the question is around novelty. Regardless, it is important that this type of work is done. Its rigour is impressive and it makes it clear that to understand how combinations of transcription factors regulate cellular properties, which then combine to shape specific behavioural outputs, remains a complex and challenging task. Here, the discussion is helpful in highlighting the importance to consider the evolutionary timeline of anatomical-developmental features, such as the conserved ground-plan of neuroblasts relative to genes such as Unc-4 having been harness in the diversification of hemilineages.

In summary, we see that the methodological and scientific rigour of this work allow the authors to interpret the data with clarity and decisiveness that would otherwise not be possible:

a) on distinguishing that only some hemilineages have a clear (anatomical and functional) requirement for Unc-4;

b) that for most that do require Unc-4 this is developmental rather than later/post-mitotic (it is not unlikely that neuron cell fate is occurring in at least two waves of specification: early spatial/temporal fate via transient cues, and then post-mitotic consolidation of fate via long-lasting gene expression programs (TF combinations). So this paper would be directly relevant to that big picture issue and this could be highlighted in copyediting with a well-crafted sentence);

c) that Unc-4 is required for very different aspects in different lineage contexts;

Together, demonstrated with such clarity, these observations help us think about how transcription factors shape nervous system development and function.

Reviewer #1 Minor Comments:

- The finding that removal of Unc-4 in post-mitotic neurons has no phenotype is very interesting; I would consider adding that finding to the Abstract and Discussion.

– I think it is important to say whether you are mentioning *larval* hemilineages or both embryonic and larval hemilineages in the Introduction and Results. For example, does the term "individual hemilineages" refer to both embryonic and post-embryonic hemilineages? For example, the last paragraph of the Introduction mentions hemilineages multiple times without distinguishing embryonic and post-embryonic hemilineages.

– Is it known whether the 7B neurons that project to the leg neuropil are direct premotor neurons? One of this lineage (A07f2) is directly premotor to Eve^+^ motor neurons in the larva (Zarin et al., BioRx55007iv 2019 Table 3) so it would be nice to add this information for the adult neurons.

– It is not clear from the text whether the Unc-4^+^ hemilineages contain only Unc-4^+^ neurons, or also contain some Unc-4^-^ neurons; please clarify.

– It is also not clear whether all cholinergic neurons are Unc-4^+^ or if some are Unc-4^-^; please clarify.

– Subsection “Unc-4 and neurotransmitter expression”, please add a citation for the statement that lin7 (NB3-2) neurons are Eve^+^. To my knowledge, the only lineages that are Eve^+^ are 1-1, 3-3, 4-2, and 7-1.

– Sometimes the Unc-4 gene is italicized and sometimes not, please be consistent.

– Could add Skeath 1998 to the reference in the second paragraph of the Introduction.

– The first paragraph of the subsection “Unc-4 function is required for proper neuronal projections in 7B neurons” says "see Materials and methods" but I can't find the data in the Materials and methods.

– The first paragraph of the subsection “Unc-4 function is required for proper neuronal projections in 7B neurons” has a broken reference that needs correction.

– I would appreciate a thin dashed line around the leg neuropil in the relevant figures.

Reviewer #2 Minor Comments:

I have only very minor points to raise:

Correct "axion projections"

Figure 2 – legend: there is some confusion with referring to panels E and F and then G and H.

Figure 7: panels C and D to replace "o" with a proper zero.

---

## [Author Response]

Reviewer #1 Minor Comments:– The finding that removal of Unc-4 in post-mitotic neurons has no phenotype is very interesting; I would consider adding that finding to the Abstract and Discussion.

We thank for the reviewer for this suggestion. As suggested, we edited the Abstract and Discussion to reflect this point as shown below.

Part of the edited Abstract:

“Through time-dependent conditional knock-out of Unc-4, we found that its function is required during development, but not in the adult, to regulate the above events.”

Part of the edited Discussion:

“Similarly, pan-neuronal deletion of Unc-4 specifically in the adult did not lead to any apparent behavioral defect even though Unc-4 expression is maintained in all Unc-4^+^ lineages throughout adult life, suggesting that Unc-4 function is dispensable in mature neurons after eclosion under standard lab conditions. Future work will be required to ascertain whether Unc-4 functions during adult life or in more than two of its expressing hemilineages during development. Nonetheless, our work shows that Unc-4 executes distinct functions in the 7B and 11A lineages.”

– I think it is important to say whether you are mentioning larval hemilineages or both embryonic and larval hemilineages in the Introduction and Results. For example, does the term "individual hemilineages" refer to both embryonic and post-embryonic hemilineages? For example, the last paragraph of the Introduction mentions hemilineages multiple times without distinguishing embryonic and post-embryonic hemilineages.

We appreciate this point and have added the following sentence to address it and clarify that our study is based entirely on postembryonic hemilineages.

Edit to the Introduction:

“In this paper, we focus only on postembryonic hemilineages, which from this point on in the paper we refer to as hemilineages for simplicity.”

– Is it known whether the 7B neurons that project to the leg neuropil are direct premotor neurons? One of this lineage (A07f2) is directly premotor to Eve^+^ motor neurons in the larva (Zarin et al., BioRxiv 2019 Table 3) so it would be nice to add this information for the adult neurons.

We are aware of the recently published Zarin et al., 2019, e*Life* paper. It is possible that larval motif is retained in the adult CNS. Unfortunately, we do not have any data for or against a premotor function of 7B neurons in the adult. Future work will investigate whether 7B neurons synapses with jump related motor neurons such as TTMn and TLMn.

– It is not clear from the text whether the Unc-4^+^ hemilineages contain only Unc-4^+^ neurons, or also contain some Unc-4^-^ neurons; please clarify.

We clarified this ambiguity by editing the Introduction as shown below:

“Unc-4 expression in these hemilineages is restricted to postmitotic neurons and it appears to mark uniformly all neurons within a hemilineage during larval and adult life (Lacin et al., 2014; Lacin and Truman, 2016).”

– It is also not clear whether all cholinergic neurons are Unc-4^+^ or if some are Unc-4^-^; please clarify.

We edited the related Result section to indicate that Unc-4 marks half the cholinergic lineages as shown below:

“We previously showed that all of the Unc-4^+^ postembryonic hemilineages are cholinergic and that half of all cholinergic hemilineages (7 of 14) express Unc-4 in the VNC (Lacinet al., 2019).”

– Subsection “Unc-4 and neurotransmitter expression”, please add a citation for the statement that lin7 (NB3-2) neurons are Eve^+^. To my knowledge, the only lineages that are Eve^+^ are 1-1, 3-3, 4-2, and 7-1.

We believe the reviewer meant 11B neurons from NB7-2. We added the references (Lacin and Truman, 2016; Lacin et al., 2019) that demonstrated Eve expression in postmitotic 11B neurons.

– Sometimes the Unc-4 gene is italicized and sometimes not, please be consistent.

We now italicized *unc-4* throughout the text.

– Could add Skeath 1998 to the reference in the second paragraph of the Introduction.

We added the indicated reference.

– The first paragraph of the subsection “Unc-4 function is required for proper neuronal projections in 7B neurons” says "see Materials and methods" but I can't find the data in the Materials and methods.

We thank to the reviewer for indicating this. We now created a subsection in the Materials and methods with a title “Lineage-specific GAL lines” to make it easier for the reader to identify these reagents.

– The first paragraph of the subsection “Unc-4 function is required for proper neuronal projections in 7B neurons” has a broken reference that needs correction.

We corrected it.

– I would appreciate a thin dashed line around the leg neuropil in the relevant figures.

Demarcating the leg neuropils is a helpful suggestion. We outlined the leg neuropils in Figure 6 as suggested.

Reviewer #2 Minor Comments:I have only very minor points to raise:Correct "axion projections"Figure 2 legend: there is some confusion with referring to panels E and F and then G and H.Figure 7: panels C and D to replace "o" with a proper zero.

We thank for the reviewer for catching these typos and errors. We have corrected all of them.